# Sensitivities to biological aerosol particle properties and ageing processes: Potential implications for aerosol-cloud interactions and optical properties

Minghui Zhang[1], Amina Khaled[1], Pierre Amato[1], Anne-Marie Delort[1], and Barbara Ervens[1]

[1]Université Clermont Auvergne, CNRS, SIGMA Clermont, Institut de Chimie de Clermont-Ferrand, 63000 Clermont-Ferrand, France

*Corresponding author*: Minghui Zhang (minghui.zhang@uca.fr)

**Abstract.**

Primary biological aerosol particles (PBAPs) such as bacteria, viruses, fungi, and pollen, represent a small fraction of the total aerosol burden. Based on process model studies, we identify trends in the relative importance of PBAP properties, e.g. number concentration, diameter, hygroscopicity, surface tension, contact angle, for their aerosol-cloud interactions and optical properties. While the number concentration of PBAPs likely does not affect total CCN concentrations globally, small changes in the hygroscopicity of submicron PBAPs might affect their CCN ability and thus their inclusion into clouds. Given that PBAPs are highly efficient atmospheric ice nuclei at T > -10 °C, we suggest that small changes in their sizes or surface properties due to chemical, physical or biological processing might translate into large impacts on ice initiation in clouds. Predicted differences in the direct interaction of PBAPs with radiation can be equally large between different species of the same PBAP type and among different PBAP types. Our study shows that not only variability of PBAP types, but also their physical, chemical, and biological ageing processes might alter their CCN and IN activities and optical properties to affect their aerosol-cloud interactions and optical properties. While these properties and processes likely affect radiative forcing only on small spatial and temporal scales, we highlight their potential importance for PBAP survival, dispersion and transport in the atmosphere.

## 1. Introduction

Although primary biological aerosol particles (PBAPs) contribute a small fraction (50 Tg yr$^{-1}$, with an upper limit of 1000 Tg yr$^{-1}$) to the total natural global aerosol emissions of ~2900-13000 Tg yr$^{-1}$ (Stocker et al., 2013), they have attracted great interest in the atmospheric science and public health community as they might affect the climate and be responsible for spreading diseases (Asadi et al., 2020; Behzad et al., 2018; Khaled et al., 2020). They consist of bacteria, proteins, viruses, fungi, pollen and other biologically-derived materials with potentially infectious, allergenic, or toxic properties (Fröhlich-Nowoisky et al., 2016).

Their mass (Graham et al., 2003; Heald and Spracklen, 2009), number concentrations (Huffman et al., 2013; Matthias-Maser et al., 1999; Forde et al., 2019a), and fractions (Jaenicke, 2005) can greatly vary depending on location (Schumacher et al., 2013; Shen et al., 2019; Wei et al., 2016; Yu et al., 2016), time of day (Kang et al., 2012) and other conditions (Graham et al., 2003; Jiaxian et al., 2019; Wu et al., 2016; Forde et al., 2019b). For example, in the Amazonian rainforest, PBAPs contribute ~20% to the mass of submicron organic aerosol (Schneider et al., 2011). In the semirural area of Mainz in central Europe, the number fraction was 1-50% for particles with diameter (D) > 0.4 µm (Jaenicke, 2005). Above the ocean, 1% of particles with 0.2 µm < D < 0.7 µm contain biological materials (Pósfai et al., 1998). Temporal variability of PBAPs was observed exhibiting peaks in the morning, during and after rain (Huffman et al., 2013; Zhang et al., 2019). To the total global PBAP emissions, bacteria contribute 0.4-1.8 Tg yr$^{-1}$, which is less than 25-31 Tg yr$^{-1}$ by fungal spores (Heald and Spracklen, 2009; Hoose et al., 2010) and 47 Tg yr$^{-1}$ by pollen (Burrows et al., 2009a, 2009b). Although the mass fraction of bacteria is small, their number concentration (~0.001-1 cm$^{-3}$) (Lighthart and Shaffer, 1995; Tong and Lighthart, 2000) is larger than that of fungal spores (~0.001-0.01 cm$^{-3}$) and pollen (~0.001 cm$^{-3}$) (Huffman et al., 2010). The concentration of viruses can reach up to ~0.1 cm$^{-3}$ in indoor air (Prussin et al., 2015) and decreases to ~0.01 cm$^{-3}$ outdoors (Després et al., 2012; Weesendorp et al., 2008). The comparably small size of viruses and bacteria (D$_{viruses}$ ~0.1 µm, D$_{bacteria}$ ~1 µm) enables relatively long residence times of several days in the atmosphere (Burrows et al., 2009a; Verreault et al., 2008).

In numerous recent review articles, it has been suggested that PBAPs can affect radiative forcing in multiple ways (*Figure 1*) (Coluzza et al., 2017; Després et al., 2012; Haddrell and Thomas, 2017; Hu et al., 2018; Šantl-Temkiv et al., 2020; Smets et al., 2016): PBAPs might directly interact with radiation by scattering or absorbing light (*Figure 1a*). While their aerosol direct effect is likely globally small due to low PBAP number concentration (Löndahl, 2014), it may be of greater interest locally and for specific wavelength ranges due to the large size of PBAPs (Myhre et al., 2013). The optical properties of PBAPs (Arakawa et al., 2003; Hu et al., 2019; Thrush et al., 2010) resemble those of other organic particles as PBAPs are largely composed of proteins and other macromolecules. Accordingly, PBAPs' optical properties can be ascribed

to specific organic functional entities such as amino groups or aromatic structures (Hill et al., 2015; Hu et al., 2019). At subsaturated relative humidity (RH) conditions, the hygroscopicity ($\kappa_{PBAP}$) determines their ability to take up water (Petters and Kreidenweis, 2007) and thus their equilibrium size, which might affect their direct radiative properties. Their hygroscopicity shows a large range ($0.03 \leq \kappa_{PBAP} \leq 0.25$), which is explained by variation of surface composition due to different types of PBAPs and/or ageing processes (Bauer et al., 2003; Haddrell and Thomas, 2017; Šantl-Temkiv et al., 2020; Sun and Ariya, 2006).

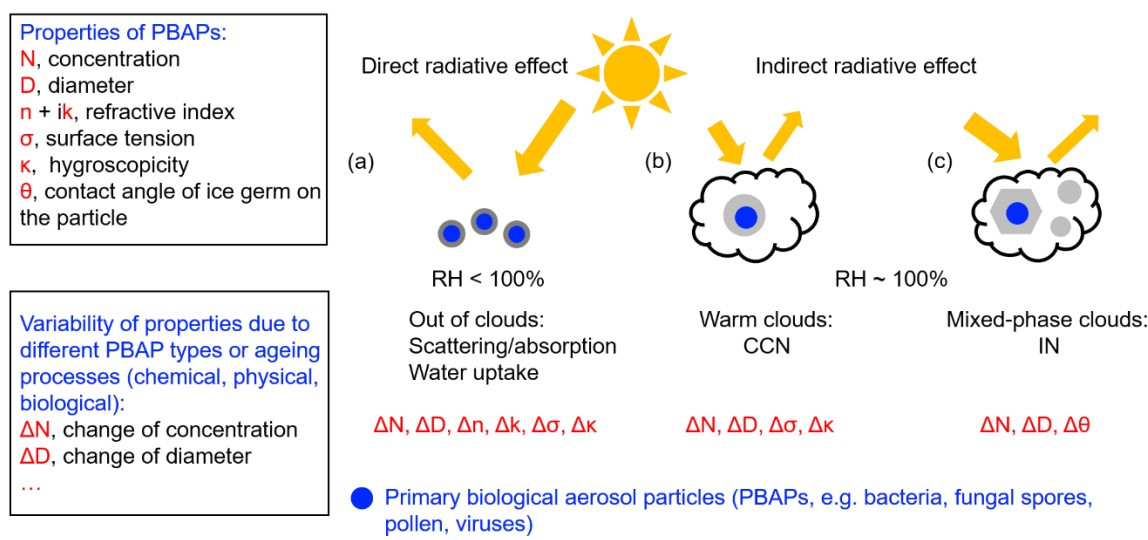

**Figure 1.** Schematic of the influence of PBAP properties and ageing processes on direct and indirect radiative effects. (a) The direct radiative forcing might be influenced by PBAP concentration ($N_{PBAP}$), diameter ($D_{PBAP}$), refractive index ($m_{PBAP} = n + ik$), surface tension ($\sigma_{PBAP}$), and hygroscopicity ($\kappa_{PBAP}$) at RH < 100%. (b) $N_{PBAP}$, $D_{PBAP}$, $\sigma_{PBAP}$, and $\kappa_{PBAP}$ might affect CCN activity and properties of warm clouds. (c) $N_{PBAP}$, $D_{PBAP}$, and contact angle of ice germ on the particle ($\theta_{PBAP}$) might affect the evolution of mixed-phase clouds.

Convective and precipitating clouds lead to efficient particle redistribution by vertical transport and removal of particles by wet deposition. Therefore, cloud-related physicochemical properties need to be constrained to determine the distribution and residence time of PBAPs in the atmosphere. Since PBAPs often have supermicron sizes, they may act as 'giant CCN' and thus induce early precipitation (Barahona et al., 2010; DeLeon-Rodriguez et al., 2013; Feingold et al., 1999). Based on a global model study, it was concluded that CCN-relevant properties need to be refined in order to further probe their role in the climate system (Konstantinidis, 2014). In particular pollen rupture leads to a huge increase in the number of subpollen particles (SPPs) (Bacsi et al., 2006; Suphioglu et al., 1992; Taylor et al., 2004; Wozniak et al., 2018). By assuming that one pollen grain releases up to $10^6$ SPPs, regional model studies suggested that the resulting SPPs can significantly suppress seasonal precipitation (Wozniak et al., 2018). Several experimental studies

have explored the CCN properties of PBAPs and determined their hygroscopicity (κ) (Ariya et al., 2009; Sun and Ariya, 2006). The role of biosurfactant production by bacteria and fungi has been also discussed in the context of their CCN activity since a lower surface tension ($\sigma_{PBAP}$) enhances water uptake (Renard et al., 2016). In addition, biosurfactant molecules that are produced by bacteria and fungi, while they reside on leaves or other surfaces, might attach to other particles, thus, increasing their CCN ability as well.

In addition to acting as CCN, some species of bacteria, fungi, and pollen can nucleate ice at high temperatures (Hoose and Möhler, 2012; Morris et al., 2004, 2008; Pouzet et al., 2017; Diehl et al., 2001, 2002), which makes them unique in terms of ice nucleation to affect the evolution of mixed-phase clouds at these temperatures (*Figure 1c*). Above vegetated forests (Tobo et al., 2013) and near the surface of the Southern Ocean (Burrows et al., 2013), PBAPs have been shown to contribute significantly to the total abundance of IN. In a high altitude mountain region of the United States, ambient measurements suggest that 16 to 76% of IN at -30 °C consist of primary biological material (Prenni et al., 2009); a similar proportion (33%) was reported at -31 to -34 °C in the Amazon basin (Pratt et al., 2009).

The radiative impacts of PBAPs, influenced by the physicochemical properties ($N_{PBAP}$, $D_{PBAP}$, $\kappa_{PBAP}$, $\sigma_{PBAP}$, $mP_{BAP}$, $\theta_{PBAP}$), summarized in *Figure 1,* can largely differ on spatial and temporal scales, leading to different conclusions regarding the climatic impacts of PBAPs (Burrows et al., 2009a, 2009b; Hoose et al., 2010; Junge and Swanson, 2008; Konstantinidis, 2014; Sahyoun et al., 2017; Sesartic et al., 2012). These properties are even more variable than represented in current models as PBAPs undergo chemical, physical and biological ageing processes (Coluzza et al., 2017; Deguillaume et al., 2008; Pöschl, 2005; Vaïtilingom et al., 2010).

- *Physical* transformations include agglomeration/fragmentation of cells (Coluzza et al., 2017; Lighthart, 1997; Zhang et al., 2019), coating with organic or inorganic components (Pöschl and Shiraiwa, 2015; Joly et al., 2015), or with solid ice or liquid water (Joly et al., 2013). These processes might alter various physicochemical properties listed in *Figure 1*. For example, the break-up of pollen or fungi due to rupture can lead to higher number concentrations by several orders of magnitude (Suphioglu et al., 1992; Wozniak et al., 2018).
- *Chemical* transformations include oxidation (Jayaraman et al., 2008; Vaïtilingom et al., 2010), nitration (Franze et al., 2005), oligomerization (Tolocka et al., 2004), degradation of macromolecules (Estillore et al., 2016), and changes of the protein conformations due to exposure to different pH (Kristinsson and Hultin, 2004). These processes lead to the modification of the protein structures and other macromolecules and thus affect PBAP optical properties (Myhre et al., 2013), CCN activity (Sun and Ariya, 2006), and IN ability (Attard et al., 2012; Kunert et al., 2019).

- *Biological* processes might be initiated by living microorganisms in PBAPs, unlike in other aerosol
  particles in the atmosphere (Amato et al., 2017; Delort et al., 2017; Joly et al., 2015). Such processes are
  generally driven by strategies to adapt to the harsh conditions in the atmosphere (e.g., rapid temperature
  and RH changes, thaw/freeze cycles, humidification/desiccation, UV exposure) (Hamilton and Lenton,
  1998; Horneck et al., 1994; Joly et al., 2015; Setlow, 2007) or to limit their atmospheric residence time
  by initiating precipitation (Hernandez and Lindow, 2019). These processes include nutrient uptake by
  biodegradation (Khaled et al., 2020), bacteria cell generation that enhances particle size and surface area
  (Ervens and Amato, 2020), formation of biofilms (extracellular polymeric substances) which enables
  PBAPs to form aggregates (Monier and Lindow, 2003, 2005; Morris et al., 2008; Sheng et al., 2010),
  expression of ice-nucleating proteins (Joly et al., 2013; Kjelleberg and Hermansson, 1984), formation of
  biosurfactants that enhances water uptake (Hernandez and Lindow, 2019; Neu, 1996), desiccation that
  decreases size of PBAPs (Barnard et al., 2013), formation of pigments (Pšenčík et al., 2004; Fong et al.,
  2001) enhancing light absorption, fungal spore germination (Ayerst, 1969) or formation of bacteria
  endospores (Enguita et al., 2003) that increases $N_{PBAP}$, and metabolism of cellular components
  (membranes, proteins, saccharides, osmolytes, etc) (Fox and Howlett, 2008; Xie et al., 2010). To date,
  the uncertainties introduced by these PBAP ageing processes in the estimate of PBAP radiative effects,
  their atmospheric residence time and distribution can only be assessed qualitatively due to the lack of
  comprehensive data. However, it may be expected that some of these ageing processes lead to similar
  differences in PBAP properties than differences between PBAP types.

In our study, we give a brief overview of the PBAP properties in *Figure 1* and summarize which *chemical,
physical* and *biological* processes may alter these properties (*Section 2*). By means of process models
(*Section 3*), we explore in a simplistic way the relative importance of these PBAP properties and ageing
processes for the effects depicted in *Figure 1* (*Section 4*). Our model sensitivity studies are set up such that
we identify trends and their relative importance to show the sensitivities to individual properties and ageing
processes that impact PBAP properties in the atmosphere. The results of our sensitivity studies allow a
ranking of the importance of the various PBAP properties and processes in terms of their aerosol-cloud
interactions and optical properties (*Section 5*). Finally, we give some guidance on the need of future
laboratory, field and model studies to more accurately describe potential radiative effects, distribution and
residence time of PBAPs in the atmosphere.

## 2. Physicochemical properties and processes of PBAPs

Literature data on physicochemical parameters of PBAPs are summarized in *Table 1*. It is not our goal to
repeat exhaustive reviews on these individual properties; for this, we refer to previous overview articles
(Bauer et al., 2003; Coluzza et al., 2017; Deguillaume et al., 2008; Després et al., 2012; Fröhlich-Nowoisky

et al., 2016; Hoose and Möhler, 2012; Huffman et al., 2020; Šantl-Temkiv et al., 2020). We rather aim at using characteristic orders of magnitude of these properties as input data to our process models (***Section 3***). Therefore, we only give a brief overview on the ranges and variability of these properties for different PBAP types and due to various ageing processes.

### 2.1 PBAP number size distribution parameters ($N_{PBAP}$ and $D_{PBAP}$)

The number concentration ($N_{PBAP}$) of most PBAP types is in the range of $0.001 \leq N_{PBAP} \leq 0.1$ cm$^{-3}$ (***Table 1***). The number concentration of bacteria is higher than that of fungal spores and pollen although the mass concentration of bacteria is lower (Burrows et al., 2009a; Heald and Spracklen, 2009; Hoose et al., 2010). $N_{PBAP}$ can vary by about three orders of magnitude among different ecosystems, locations, seasons, and time of the day (Huffman et al., 2010, 2020; Matthias-Maser et al., 2000a, 2000b; Schumacher et al., 2013). The PBAP diameter ($D_{PBAP}$) covers a broad range of $0.01$ μm $\leq D_{PBAP} \leq 100$ μm. This parameter usually refers to the mass equivalent diameter, which is the diameter of a sphere with the same mass as a non-spherical PBAP. The size depends on the type of PBAPs, and on changes due to biological and physical processing. Viruses are reported to be the smallest PBAPs ($0.01$ μm $\leq D_{viruses} \leq 0.3$ μm) while pollen is the largest ($5$ μm $\leq D_{pollen} \leq 100$ μm) (***Table 1***). Biological processing, such as cell generation, might increase the size of particles by producing secondary biological aerosol mass (Ervens and Amato, 2020; Sattler et al., 2001). Typical bacterial cell generation rates are in the range of $0.1$ to $0.9$ h$^{-1}$ (Ervens and Amato, 2020). Efficient generation in the atmosphere is assumed to be largely restricted to the time of cell exposure to liquid water (i.e., in-cloud). With an average atmospheric residence time of ~1 week (Burrows et al., 2009b) and an average in-cloud time fraction of ~15% (Lelieveld and Crutzen, 1990), it can be estimated that the generation time scale of bacteria cells in the atmosphere is on the order of ~20 h. Thus, for example, $D_{bacteria}$ may increase from 1 μm to 2 μm after one week in the atmosphere assuming a generation rate of $0.3$ h$^{-1}$. Other rates, such as the cell growth, are usually much smaller (Marr, 1991; Middelboe, 2000; Price and Sowers, 2004; Sattler et al., 2001; Vrede et al., 2002), and thus, contribute less efficiently to a change in $D_{PBAP}$. In addition, the formation of extracellular polymeric substances might lead to the formation of biofilms, which increase PBAP size by forming agglomerates (Monier and Lindow, 2003, 2005). Agglomerate formation might be also described as a physical process, when PBAPs (e.g. bacteria) attach to other particles (e.g. dust) (Després et al., 2012; Lighthart, 1997), which can result in particle sizes on the order of ~10 μm. At high RH and during precipitation or thunderstorms, pollen absorb water and one pollen grain can release ~$10^3$ SPPs due to osmotic pressure (Grote et al., 2001; Suphioglu et al., 1992). This process can result in fragments with diameter of 1-4 μm and number concentrations of $N_{SPP}$ ~$0.1$ cm$^{-3}$ during thunderstorms (Zhang et al., 2019). These concentrations correspond to ~1 to 25 ng m$^{-3}$ ($D_{SPP} < 2$ μm) (Miguel et al., 2006). Laboratory chamber measurements have shown that SPPs from rupture of fresh birch pollen or grass pollen have

diameters of in the range of 0.03 to 4.7 µm (Taylor et al., 2002, 2004). Recent laboratory measurements suggest that also fungal spores can rupture, resulting in subfungi particles (SFPs) with $D_{SFP}$ of 0.03 to 0.9 µm after exposure to high relative humidity (China et al., 2016). Ambient measurements suggest $N_{SFP}$ of 150 to 455 cm$^{-3}$ (10 nm < $D_{SFP}$ < 100 nm) after rainfall; observed peaks in aerosol size distributions at 20 nm < $D_{SFP}$ < 50 nm which frequently appeared 1.5 days after rain events were ascribed to such rupture events (Lawler et al., 2020).

## 2.2 Optical properties of PBAPs: Complex refractive index ($m_{PBAP} = n + ik$)

The scattering and absorption of particles are commonly described by the refractive index $m_{PBAP}$ with real part ($n_{PBAP}$) and imaginary parts ($k_{PBAP}$) that depend on the chemical composition and wavelength of irradiation. Arakawa et al. (2003) reported $1.5 \leq n_{PBAP} \leq 1.56$ and $3 \cdot 10^{-5} \leq k_{PBAP} \leq 6 \cdot 10^{-4}$ for bacteria (*Erwinia herbicola*) in the wavelength range of 0.3-2.5 µm. Other groups found a broader range of n and k (*Table 1*) for different PBAPs and irradiation wavelengths (Hu et al., 2019; Thrush et al., 2010). The imaginary part can vary by three orders of magnitude for different PBAP types (Hu et al., 2019). Hill et al. (2015) showed that the refractive index of PBAPs can be estimated based on the chemical composition. They reported 1.59 + i0.045 for *Bacillus vegetative* cells at 0.266 µm. Also PBAP shape (e.g. core-shell structure, hexagonal grids, and barbs), as it has been demonstrated for pollen, influences the optical properties (Liu and Yin, 2016). Due to the similarity of the molecular structure of organic macromolecules (e.g. proteins) and secondary organic aerosols (SOA), it can be likely assumed that nitration might alter the PBAP refractive index similar to that of SOA. Experimental results show $1.528 \leq n \leq 1.576$ and $0 \leq k \leq 0.02$ for fresh SOA in the wavelength range of 0.3-0.56 µm; after nitration, the real part increases to $1.549 \leq n \leq 1.594$ and the imaginary part increases to $0.0002 \leq k \leq 0.04$ (Liu et al., 2015).

**Table 1.** Physicochemical properties of various types of PBAPs and their changes due to physical, chemical and biological ageing processes based on literature data.

| PBAP types | Physicochemical properties | | | | | | |
|---|---|---|---|---|---|---|---|
| | Concentration N (cm$^{-3}$) | Diameter D (µm) | Complex refractive index m (λ) = n + ik | Hygroscopicity κ | Surface tension σ (mN m$^{-1}$) | Number fraction of PBAPs with IN active molecules | Contact angle θ (°) |
| Bacteria | 0.001-1 (1) | 1 (17) 0.6-7 (18) | n: 1.5-1.56, k: $3 \cdot 10^{-5}$-$6 \cdot 10^{-4}$ (24); n: 1.5-1.56, k: 0-0.04 (25); n: 1.25-1.87, k: 0-0.5 (26) | 0.11-0.25 (27) | 25, 30, 55, 72 (35) | ~0.1%, ~1%, ~10% (36) | 32-34 (39); 4-20 (40); 28, 33, 44 (41) |

| | | | | | | |
|---|---|---|---|---|---|---|
| Fungal spores | 0.001-0.01 (2) | 3-5 (4); 1-30 (5); | n: 1.25-1.75, k: 0-0.32 (26) | | | 30-33 (42) |
| Subfungi particles (SFPs) | 150-455 (3) | 0.01-0.1 (3); 0.02-0.05 (3); 0.03-0.9 (19) | | | | |
| Fern spores | $10^{-5}$ (4) | 1-30 (4) | | | | |
| Pollen | 0.001 (5) | 5-100 (20) | n:1.3-1.75, k: 0.01-0.2 (26) | 0.03-0.073 (28); 0.036-0.048 (29); 0.05-0.1 (30); 0.08-0.17 (31) | ~100% (37,38) | 14-30 (40); 15, 16.3 (43) |
| Subpollen particles (SPPs) | 0.1 (6) | 1-4 (6); 0.03-4 (21); 0.12-4.67 (22) | | 0.14-0.24 (32); 0.12-0.13 (33); 0.1-0.2 (34) | | |
| Viruses | 0.01 (4) | 0.01-0.3 (4); 0.04-0.2 (23) | | | | |
| Ambient PBAPs | 0.1-1 (7); 1-8 (8) | > 0.4 (7,8) | | | | |
| Ambient PBAPs | 0.2-1.2 (9); 0.04-0.13 (10); 0.012-0.095 (11); 0.01-1.4 (12); 0.57-3.3 (13); 0.1-0.43 (14); 0.02-0.09 (15); 0.005-0.5 (16) | > 1 (9-16) | | | | |

| **Ageing processes of PBAPs** | | | |
|---|---|---|---|
| | Physical ageing | Chemical ageing | Biological ageing |
| Bacteria | Agglomeration: $\Delta D > 0$, $\Delta N < 0$ (18) | Nitration: $\Delta n > 0$, $\Delta k > 0$ (44); Nitration: $\Delta\theta \sim 1°$ (41); pH changes: $\Delta\theta \sim 1.5°$ (41). | Biosurfactant production: $\sigma < 0$ (35); Biofilm formation: $\Delta D > 0$ (45); Endospore formation: $\Delta N > 0$ (46); Cell generation: $\Delta D > 0$ (47); Desiccation: $\Delta D < 0$ (48); Pigment formation : $\Delta k > 0$ (49,50); IN protein expression: $\Delta\theta < 0$ (no data yet) |
| Fungi | Rupture: $\Delta D < 0$, $\Delta N > 0$ (3,19) | | Biosurfactant production: $\sigma < 0$ (35); Germination: $\Delta N > 0$ (49); Desiccation: $\Delta D_{BAP} < 0$ (48) |
| Pollen | Rupture: $\Delta D < 0$, $\Delta N > 0$ (6,21,22) | Oxidation: $0.5 \leq \Delta\theta \leq 0.8°$ (43) | |

(1) Total bacteria, Tong and Lighthart et al., 1999; (2) Elbert et al., 2007; (3) After rainfall, Lawler et al., 2020; (4) Després et al., 2012; (5) blooming times, Huffman et al. 2010; (6) thunderstorm times, Zhang et al., 2019; (7) Based on protein dyes, Lake Baikal, Russia, Jaenicke, 2005; (8) Based on protein dyes, Mainz, Germany, Jaenicke, 2005; (9) In the Amazon, Whitehead et al., 2016; (10) In the Amazon, Huffman et al., 2012; (11) Puy de Dôme,  Gabey et al. 2013; (12) In megacity Beijing, China, Wei et al., 2016;

(13) In Megacity Nanjing, China, Yu et al., 2016; (14) High altitude, Ziemba et al., 2016; (15) High altitude, Perring et al. 2015; (16) High concentration observed during and after rain, Huffman et al., 2013; (9) to (16) are based on autofluorescence of PBAPs; (17) Burrows et al., 2009a; (18) Lighthart 1997; (19) China et al., 2016; (20) Pöhlker et al., 2013; (21) Taylor et al., 2004; (22) Taylor et al., 2002; (23) Verreault et al., 2008; (24) Arakawa et al., 2003; (25) Thrush et al., 2010; (26) Hu et al. 2019; (27) Lee et al., 2002; (28) Pope et al. 2010; (29) Tang et al., 2019; (30) Chen et al., 2019; (31) Griffiths et al., 2012 ; (32) pollenkitt, Prisle et al., 2019; (33) Mikhailov et al., 2019; (34) Mikhailov et al., 2020; (35) Renard et al., 2016; (36) T ~-10 ℃, immersion freezing, *Pseudomonas syringae* bacteria, *Pseudoxanthomonas* sp., *Xanthomonas* sp., Joly et al., 2013; (37) deposition freezing for pollen, Diehl et al., 2001; (38) immersion and contact freezing for pollen, Diehl et al., 2002; (39) Hoose and Möhler, 2012; (40) Chen et al., 2008; (41) immersion freezing for *Pseudomonas syringae*, and *Pseudomonas fluorescens*, Attard et al., 2012; (42) immersion freezing for fungi, Kunert et al., 2019; (43) deposition freezing of silver birch and grey alder pollen, Gute and Abbatt, 2018; (44) nitrated SOA (toluene as precursor) to represent nitrated BAP, Liu et al., 2015; (45) Morris et al., 2008; (46) Enguita et al., 2003; (47) Ervens and Amato, 2020; (48) Barnard et al., 2013; (49) Pšenčík et al., 2004; (50) Fong et al., 2001.

## 2.3 PBAP Properties relevant for CCN activation

### 2.3.1 Hygroscopicity ($\kappa_{PBAP}$) of PBAPs

The hygroscopicity determines the PBAP hygroscopic growth factor (gf, as the ratio of wet to dry particle diameter) at subsaturated RH conditions and their CCN activity; it is usually expressed as the hygroscopicity parameter $\kappa$ (Petters and Kreidenweis, 2007). Lee et al. (2002) reported gf = 1.16 for *Bacillus subtilis* and gf = 1.34 for *Escherichia coli* at RH ~85%. Based on these growth factors, $\kappa_{bacteria} = 0.11$ and $\kappa_{bacteria} = 0.25$ for these bacteria can be calculated. The hygroscopicity of pollen is similar to that of bacteria: The $\kappa$ value of intact pollen grains falls into the range of $0.03 \leq \kappa_{pollen} \leq 0.17$ (Chen et al., 2019; Pope, 2010; Tang et al., 2019), pollenkitts (which are parts of pollen surface) and SPPs (which are fragments after rupture) are slightly more hygroscopic ($0.14 \leq \kappa_{pollenkitt} \leq 0.24$, $0.1 \leq \kappa_{SPP} \leq 0.2$) (Mikhailov et al., 2019; Prisle et al., 2019; Mikhailov et al., 2020) than intact pollen grains, which can be explained by the nonuniform composition of pollen (Campos et al., 2008).

### 2.3.2 Surface tension ($\sigma_{PBAP}$) of PBAPs

In most model studies that explore CCN activation, it is assumed that particles have a surface tension close to that of water ($\sigma_{water}$ = 72 mN m$^{-1}$). This assumption is likely justified under many conditions due to the strong dilution of internally mixed aerosol particles near droplet activation. There are numerous studies that postulate that surfactants in aerosol particles might influence the surface tension sufficiently to significantly change their CCN activity (Bzdek et al., 2020; Facchini et al., 1999; Lowe et al., 2019; Nozière et al., 2014). These surfactants are usually assumed to have natural sources such as the ocean surface (Gérard et al., 2019; Ovadnevaite et al., 2017). Another source of surfactants might be living microorganisms that produce biosurfactants which enhance surface hygroscopicity and decrease surface tension (Akbari et al., 2018). These biosurfactants might not only be associated with PBAPs themselves as they are deposited on surfaces (e.g. leaves) where they can be taken up by other particles. Renard et al. (2016) reported that 41% of tested strains actively produce surfactant with $\sigma_{PBAP}$ < 55 mN m$^{-1}$ and 7% of tested strains can produce extremely efficient biosurfactants with $\sigma_{PBAP}$ < 30 mN m$^{-1}$. All of these tested strains were collected and isolated in

cloud water samples. The most efficient biosurfactants ($\sigma_{PBAP}$ < 45 mN m$^{-1}$) are produced by *Pseudomonas* and *Xanthomonas* bacteria (78%) and *Udeniomyces* fungi (11%). For these biosurfactants, we fit the following linear approximation based on the experimental data:

$$\sigma_{PBAP} = 89.6 - 2.9 \cdot C_{biosurf} \qquad \text{if } 6 \text{ mg L}^{-1} \leq C_{biosurf} \leq 22 \text{ mg L}^{-1} \qquad (1)$$

where $\sigma_{PBAP}$ is PBAP surface tension in (mN m$^{-1}$) and $C_{biosurf}$ is the biosurfactant concentration in (mg L$^{-1}$). Higher and lower biosurfactant concentrations may be approximated with 25 mN m$^{-1}$ and 72 mN m$^{-1}$ for simplicity. *Equation 1* implies that the concentration of biosurfactant on the surface is the same as in the bulk. Recent studies suggest that the surface concentration of surfactants is higher than the bulk concentration (Bzdek et al., 2020; Lowe et al., 2019; Ruehl et al., 2016). Thus, a smaller amount of biosurfactants ('critical micelle concentration') than suggested by *Equation 1* might be sufficient to significantly decrease $\sigma_{PBAP}$. The biosurfactant concentration depends both on the dilution (amount of water) and on the mass fraction of biosurfactants in the particle. The mass fraction has not been determined for biosurfactants; however, other surfactants have been shown to contribute ~0.1% to the total particle mass (Gérard et al., 2019).

### 2.4 PBAP properties relevant for ice nucleation

### 2.4.1 Number fraction of PBAPs with IN active macromolecules

In freezing experiments of pollen, it has been demonstrated that all particles freeze at sufficiently low temperatures, i.e. the number fraction of PBAPs that have IN active molecules can be assumed as ~100%. Both condensation and immersion/contact freezing led to frozen fractions of 100% at T = -18 °C (Diehl et al., 2001) and T = -20 °C (Diehl et al., 2002), respectively. However, it has been shown that bacteria of the same species and within the same population often exhibit different ice nucleation behavior (Bowers et al., 2009; Failor et al., 2017; Fall and Fall, 1998; Lindow et al., 1978; Morris et al., 2004). This behavior has been explained by various expression levels of IN-active macromolecules that are located at the cell surface. Under conditions such as phosphate starvation, the expression level might be higher, which is a strategy to reach nutrients after destroying the cells of plants by freezing (Fall and Fall, 1998). For example, only 0.1 to 10% of *Pseudomonas syringae* cells express IN active macromolecules (Joly et al., 2013). Bacteria from the same population without expression of such molecules did not freeze under the experimental conditions.

### 2.4.2 Contact angle between substrate and ice ($\theta_{PBAP}$)

In agreement with previous studies, we base our discussion on the contact angle as a fitting parameter in the classical nucleation theory (CNT) to parametrize the frozen fraction observed in experiments. If not reported in the respective experimental studies, we assumed a freezing time of 10 seconds to derive θ from experimental data, in agreement with many experimental conditions (Attard et al., 2012; Gute and Abbatt,

2018; Kunert et al., 2019). All CNT model equations and parameters are identical to those as described by Ervens and Feingold (2012); Hoose and Möhler (2012) discussed different assumptions made for the various variables in the CNT in previous ice nucleation studies. Chen et al. (2008) reported $4° \leq \theta_{bacteria} \leq 20°$ and $14° \leq \theta_{pollen} \leq 30°$. Based on the measurements by Attard et al. (2012), we derived values of 28°, 33°, and 44° for different species of bacteria. $\theta$ values for fungi based on the measurements by Kunert et al. (2019) are similar ($30° \leq \theta_{fungi} \leq 33°$). Gute and Abbatt (2018) performed deposition freezing experiments of pollen; based on their experiments, we fitted $\theta_{pollen} = 15°$ for silver birch and $\theta_{pollen} = 16.3°$ for grey alder. Hoose and Möhler (2012) reported the ice nucleation active surface site (INAS) density of various bacteria at -5 °C ($10^{2.5}$-$10^{10}\,m^{-2}$). Using CNT, we fitted a contact angle to their data, resulting in the range of $32° \leq \theta_{bacteria} \leq 34°$.

Chemical processes (e.g. nitration) can change the molecular surface of PBAPs by e.g., adding nitro groups to tyrosine residues of proteins (Estillore et al., 2016), which can alter the IN activity. Attard et al. (2012) measured the cumulative fraction of IN among a population of bacteria before and after nitration for 16-18 h. The residence time of aerosol particles in the atmosphere is from hours to weeks, which means that the experimental nitration times might be a realistic time scale. Based on these data, we calculated that the contact angle increased by ~1° after nitration for some bacteria. In contrast, Kunert et al. (2019) reported that protein nitration does not influence the cumulative fraction of IN for 65 species of fungi investigated. In order to study the oxidation effect, Gute and Abbatt (2018) exposed pollen to OH radicals and measured the cumulative frozen fraction of pollen in terms of deposition freezing. We calculated that the contact angle increased by ~$0.5° \leq \Delta\theta_{pollen} \leq 0.8°$ after oxidation. While experimental conditions are often optimized so that a large fraction of particles become nitrated or oxidized, only a small fraction of ambient proteins (~0.1%) have been found to be nitrated (Franze et al., 2005). In addition, Attard et al., (2012) showed that a decrease of pH from 7.0 to 4.1, led to a decrease of the cumulative fraction of IN of *P. syringae* (32b-74) from $10^{-2}$ to $10^{-8}$ at T = -4 °C. This change can be described by an increase of $\theta$ from 28.7° to 30.3° ($\Delta\theta_{bacteria}$ ~1.6°). *P. syringae* (CC0242), Snomax®, and *P. fluorescens* exhibited similar increases of $\Delta\theta_{bacteria}$ ~ 1.5° for the same change in pH.

### 3. Model description

### 3.1 Box model: Scattering/absorption of wet particles at RH < 100% calculated by Mie theory

A box model was used to calculate total scattering/absorption based on Mie theory (Bohren, 1983) for a constant aerosol distribution at different RH. Water uptake by particles is calculated based on Köhler theory. Mie theory is applied to calculate total scattering and absorption of the wet aerosol population as a function of D, N, and m at different wavelengths ($\lambda$). The input aerosol size distribution is based on ambient measurements by an ultraviolet aerodynamic particle sizer (UV-APS) in central Europe (Zhang et al., 2019)

that cannot detect particles with D < 0.5 μm. At $\lambda \geq 300$ nm, the particles with D > 3 μm interact with light

by geometric scattering, rather than Mie scattering. Therefore, we only consider particles with diameters of 0.5 μm < D < 2.8 μm in 24 size classes to represent ambient aerosol particles relevant for our study with a concentration of $N_{other,\ S(opt)} = 1.4$ cm$^{-3}$. Thus, the simulations focus on PBAPs in this size range and exclude smaller (e.g. viruses, SFPs or SPPs) and larger (e.g. pollen grains) particles. We consider one additional PBAP size class in the model with specific parameters ($N_{PBAP}$, $D_{PBAP}$, $m_{PBAP}$, $\kappa_{PBAP}$, $\sigma_{PBAP}$).

Calculations are performed for RH of 10% and 90%, i.e. for different PBAP growth factors. In a series of sensitivity studies of optical simulations ($S_{opt1}$ to $S_{opt13}$; ***Table 2***), we explore the sensitivity of scattering and absorption to $N_{PBAP}$, $D_{PBAP}$, $\kappa_{PBAP}$, and $m_{PBAP}$ ($m_{PBAP} = n + ik$). We not only compare model results for properties representing different PBAP types (e.g. $D_{bacteria}$ vs $D_{fungal}$), but also explore the ranges of property variation due to ageing processes of individual PBAP types e.g., the potential increase of bacteria diameter

($\Delta D$) due to cell generation.

***Table 2.*** Model sensitivity studies assume different physicochemical BAP parameters to investigate their effect on the optical properties (***Section 3.1***), CCN activation (***Section 3.2***) and ice nucleation (***Section 3.3***).

| Scattering/Absorption: 0.5 μm < $D_{other,\ S(opt)}$ < 2.8 μm; $N_{other,\ S(opt)} = 1.4$ cm$^{-3}$; $\kappa_{other,\ S(opt)}$: 0.3. $\sigma_{PBAP} = \sigma_{other,\ S(opt)} = 72$ mN m$^{-1}$ Composition of other particles: 90% ammonium sulfate + 10% soot | | | | | |
|---|---|---|---|---|---|
| Simulation | $N_{PBAP}$ (cm$^{-3}$) | $D_{PBAP}$ (μm) | $n(\lambda)_{PBAP}$; $k(\lambda)_{PBAP}$ | RH | $\kappa_{PBAP}$ |
| $S_{opt1}$ | 0 | - | - | 10% | - |
| $S_{opt2}$ | 0.01 | 1 | *E. herbicola:* 1.5-1.56; $3\cdot10^{-5}$-$6\cdot10^{-4}$ | | 0.25 |
| $S_{opt3}$ | 0.1 | | | | |
| $S_{opt4}$ | 1 | | | | |
| $S_{opt5}$ | 0.1 | 2 | | | |
| $S_{opt6}$ | | 3 | | | |
| $S_{opt7}$ | 1 | 2 | | | 0.03 |
| $S_{opt8}$ | | | | | 0.25 |
| $S_{opt9}$ | | | | 90% | 0.03 |
| $S_{opt10}$ | | | | | 0.25 |
| $S_{opt11}$ | | | *B subtilis*: 1.25-1.6; 0.001-0.1 | 10% | |
| $S_{opt12}$ | | | Fresh PBAPs: 1.528-1.576; 0-0.02 | | |
| $S_{opt13}$ | | | Nitrated PBAPs: 1.549-1.594; 0.0002-0.04 | | |
| **Cloud condensation nuclei (CCN):** 5 nm < $D_{other,\ S(CCN)}$ < 7.7 μm; $N_{other,\ S(CCN)} = 902$ cm$^{-3}$ | | | | | |
| | $D_{PBAP}$ (μm) | Hygroscopicity $\kappa_{PBAP}$ | | Surface tension $\sigma_{PBAP}$ (mN m$^{-1}$) | |
| $S_{CCN1}$ | 0.5 | 0.25 | | 72 | |
| $S_{CCN2}$ | | 0.03 | | | |
| $S_{CCN3}$ | 0.1 | 0.25 | | | |
| $S_{CCN4}$ | | 0.03 | | | |
| $S_{CCN5}$ | 0.05 | 0.25 | | | |
| $S_{CCN6}$ | | 0.03 | | | |
| $S_{CCN7}$ | 0.5 | 0.03 | | 25 | |
| $S_{CCN8}$ | 0.1 | | | | |

| | | |
|---|---|
| S$_{CCN9}$ | 0.05 |

**Ice nuclei (IN):**
46nm < D$_{other, S(IN)}$ < 2.5 µm; N$_{other, S(IN)}$ = 100 cm$^{-3}$
N$_{\textbf{PBAP, IN}}$ / N$_{\textbf{PBAP}}$ = 10%
θ$_{other, S(IN)}$: 80°

| | N$_{PBAP}$ (cm$^{-3}$) | D$_{PBAP}$ (µm) | Contact angle (θ$_{PBAP}$) of ice germ | Cloud base temperature (°C) |
|---|---|---|---|---|
| S$_{IN1}$ | 1 | 1 | 37° | -8 |
| S$_{IN2}$ | 0.01 | | | |
| S$_{IN3}$ | 0.001 | | | |
| S$_{IN4}$ | 0.01 | 2 | | |
| S$_{IN5}$ | | 5 | | |
| S$_{IN6}$ | | 1 | 4° | -5.5 |
| S$_{IN7}$ | | | 20° | -6 |
| S$_{IN8}$ | | | 40° | -9 |
| S$_{IN9}$ | | | 38° | -8 |

### 3.2 Adiabatic parcel model

#### 3.2.1 CCN activation in warm clouds

An adiabatic parcel model was applied to simulate the formation of warm clouds (Ervens et al., 2005; Feingold and Heymsfield, 1992). The activation of an aerosol population to cloud droplets is described as a function of N, D, κ, and σ. The dry aerosol size distribution covers a size range of 5 nm < D$_{other, S(CCN)}$ < 7.7 µm with N$_{other, S(CCN)}$ = 902 cm$^{-3}$, as being typical for moderately polluted continental conditions. Similar to the studies on optical properties (*Section 3.1*), we assume that one aerosol size class is composed of biological material, for which we vary D$_{PBAP}$, κ$_{PBAP}$, and σ$_{PBAP}$ to explore the role of differences in PBAP types and ageing processes on cloud droplet activation with CCN simulations (S$_{CCN1}$ to S$_{CCN9}$, *Table 2*).

#### 3.2.2 Ice nucleation in mixed-phase clouds

The adiabatic parcel model as used for the CCN calculations was extended by the description of immersion freezing based on classical nucleation theory (Ervens et al., 2011). At each model time step (1 second), the frozen fraction of PBAPs is calculated; if 1% or more of the IN size class are predicted to freeze in a given time step, a new size class of ice particles is generated in the model, for which ice growth is described. We consider an aerosol size distribution with 46 nm < D$_{other, S(IN)}$ < 2.48 µm in nine size classes and N$_{other, S(IN)}$ = 100 cm$^{-3}$, as found in Arctic mixed-phase clouds. The aerosol population includes one additional PBAP size class, which is the only one that includes potentially freezing IN under the model conditions. Similar to the analysis by Ervens et al. (2011), we compare the evolution of the ice liquid water contents (IWC and LWC) expressed in mass fractions [%] whereas 100% corresponds to the total water (ice + liquid + vapor) mixing ratio that is constant under the adiabatic model conditions. Input values of D$_{PBAP}$, N$_{PBAP}$, and θ$_{PBAP}$ are varied in IN simulations (S$_{IN1}$ to S$_{IN9}$) (*Table 2*).

## 4. Results and discussion

### 4.1 Sensitivity of optical properties at subsaturated conditions (RH < 100%) to PBAP properties

#### 4.1.1 Influence of concentration ($N_{PBAP}$) and diameter ($D_{PBAP}$) on scattering and absorption

As explained in ***Section 3.1***, in the sensitivity studies of optical properties, we consider only particles with D in the same range as $\lambda$ so that scattering and absorption can be calculated by Mie theory. In ***Figure 2***, we compare the total scattering coefficient for a case without PBAPs ($S_{opt1}$) to that predicted for $N_{PBAP} = 0.01$ cm$^{-3}$ ($S_{opt2}$), $N_{PBAP} = 0.1$ cm$^{-3}$ ($S_{opt3}$) and $N_{PBAP} = 1$ cm$^{-3}$ ($S_{opt4}$). At $N_{PBAP} = 0.01$ cm$^{-3}$, the effect on total scattering coefficient is negligible. At $N_{PBAP} = 0.1$ cm$^{-3}$ the total scattering coefficient increases by 15% to 18% in the range of 0.3 µm $\leq \lambda \leq 1.5$ µm although the number fraction of PBAPs is only 6%. At a higher concentration ($N_{PBAP} = 1$ cm$^{-3}$), the total scattering coefficient changes by a factor of 0.5 to 2 depending on $\lambda$. Note that the atmospheric concentration of other particles ($N_{other, S(opt)}$) might be higher than used in the above model (1.4 cm$^{-3}$); therefore, the predicted increase of scattering coefficient is likely an overestimate. The absorption coefficient of the total aerosol population does not change (***Figure S1***).

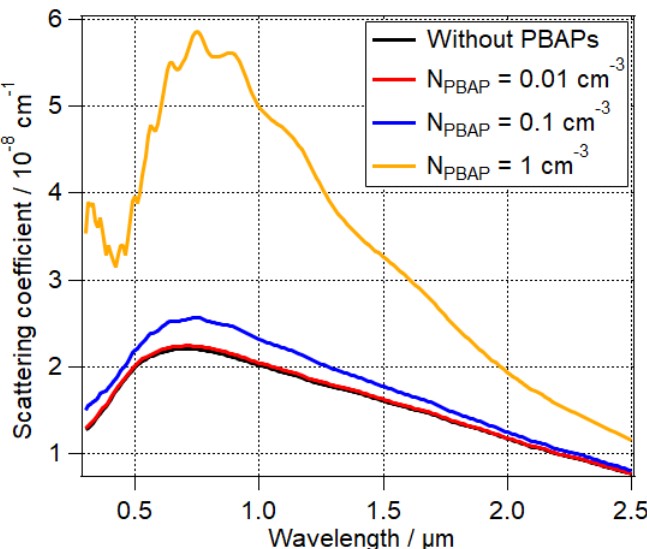

**Figure 2.** Total scattering coefficient for different PBAP number concentrations. The detailed input parameters can be found in ***Table 2***. The black, red, blue, and brown lines correspond to $S_{opt1}$, $S_{opt2}$, $S_{opt3}$, and $S_{opt4}$ in ***Table 2***, respectively.

$D_{PBAP}$ also affects the scattering coefficient of the aerosol population significantly (***Figure 3***). $D_{PBAP} = 1$ µm ($S_{opt3}$) and $D_{PBAP} = 2$ µm ($S_{opt5}$) can be considered to represent different PBAP types such as bacteria and fungi, respectively, or an aged bacteria cell that has undergone processing by cell generation (Ervens and Amato, 2020). For these assumptions, the scattering coefficient increases depending on $\lambda$, with the largest changes of 73% to 100% at $\lambda > 1.5$ µm when $D_{PBAP}$ increases from 1 µm to 2 µm ($S_{opt5}$). Larger

PBAPs ($D_{PBAP}$ = 3 µm, $S_{opt6}$) such as SPPs and fungal spores lead to an increase in the scattering coefficient by a factor of 1.4 to 4.7 depending on λ. The absorption coefficient of the aerosol population remains nearly the same (***Figure S2***).

The results in ***Figure 3*** clearly show that the size of PBAPs needs to be known in order to assess their optical properties. Even a relatively small variation in particle diameter from 1 to 2 µm due to different types or to cell diameter changes ($\Delta D_{PBAP}$) might lead to change in scattering coefficient by 8-100 % depending on λ. Given that the diameter ($D_{PBAP}$) might vary by four orders of magnitude among different PBAP types, our analysis shows that different sizes for the various PBAP types need to be taken into account when their optical properties are evaluated.

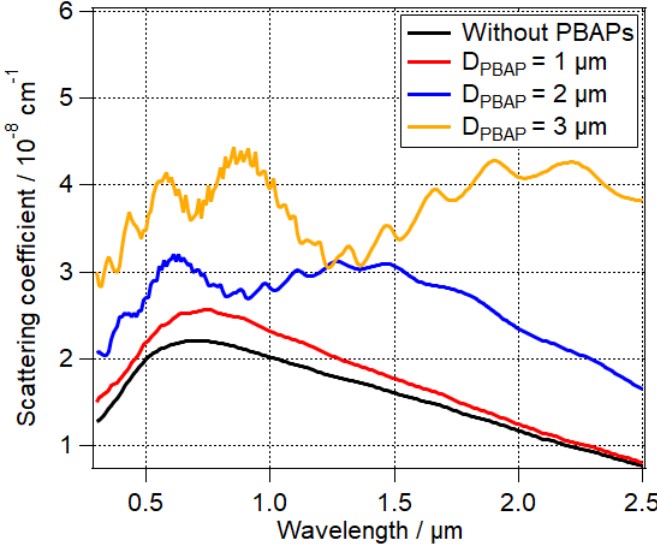

**Figure 3.** Total scattering coefficient for different PBAP diameters. The detailed input parameters can be found in ***Table 2***. The black, red, blue, and brown lines correspond to $S_{opt1}$, $S_{opt3}$, $S_{opt5}$, and $S_{opt6}$, respectively.

In our model studies, we make the simplistic assumption of spherical PBAP particles. Microscopic imaging has shown that aerosol particles are not spherical but exhibit a variety of different shapes (Valsan et al., 2015; Wittmaack et al., 2005; O'Shea et al., 2019). The consequences of the assumptions of spherical versus non-spherical pollen on the derivation of optical properties at a wavelength of 0.65 µm have been recently discussed (Liu and Yin, 2016). The extinction efficiency (sum of scattering efficiency and absorption efficiency) can vary by a factor of one to three for small pollen with D < 4 µm. For larger pollen with D > 5 µm, the extinction efficiency varies by ~25% (Liu and Yin, 2016). Non-sphericity of particles might translate into the same changes as caused by different particles sizes, which might induce uncertainties including optical depth and surface albedo (Kahnert et al., 2007). These uncertainties on scattering and

absorption caused by non-spherical shape might be of comparable magnitude to that caused by the complex refractive index (Yi et al., 2011).

**4.1.2 Influence of hygroscopicity ($\kappa_{PBAP}$) and surface tension ($\sigma_{PBAP}$) on scattering and absorption**

As discussed in ***Section 2.3***, the growth factor ($gf_{PBAP}$) might vary depending on PBAP hygroscopicity ($\kappa_{PBAP}$) and surface tension ($\sigma_{PBAP}$). ***Figure 4*** shows the influence of $\kappa$ on scattering and absorption at RH of 10% ($S_{opt7}$, $S_{opt8}$) and 90% ($S_{opt9}$, $S_{opt10}$). At RH = 10% ($S_{opt7}$, $S_{opt8}$), the influence of PBAPs on scattering coefficient of total particles is small (< 19%) and the influence on absorption coefficient is negligible. At high RH = 90%, the water content of particles is significantly higher when $\kappa = 0.25$ as compared to $\kappa = 0.03$.

Assuming $\kappa = 0.25$ ($S_{opt10}$) instead of $\kappa = 0.03$ ($S_{opt9}$), leads to an increase of the scattering coefficient by 17 to 90% at RH = 90%. Also the absorption coefficient increases by ~40% at $\lambda > 2$ µm. This trend can be explained as the imaginary part of water is higher by three orders of magnitude at $\lambda \sim 2$ µm compared to that at $\lambda \sim 1$ µm (Kou et al., 1993). It can be concluded that the importance of $\kappa_{PBAP}$ increases at higher RH, as under these conditions PBAP hygroscopic growth is most efficient.

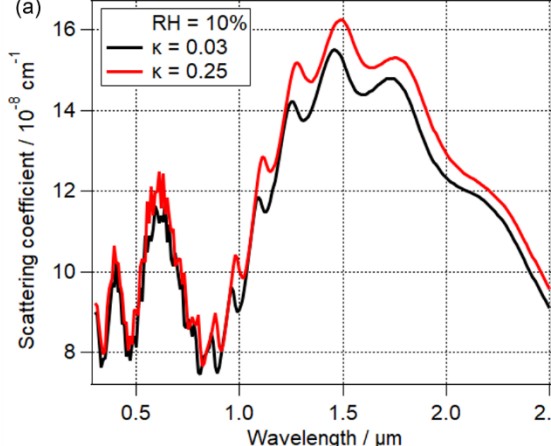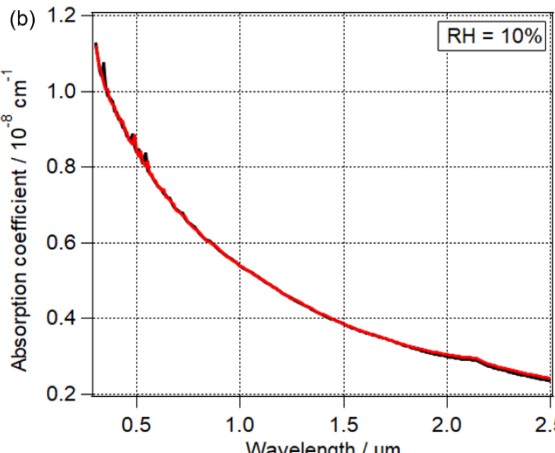

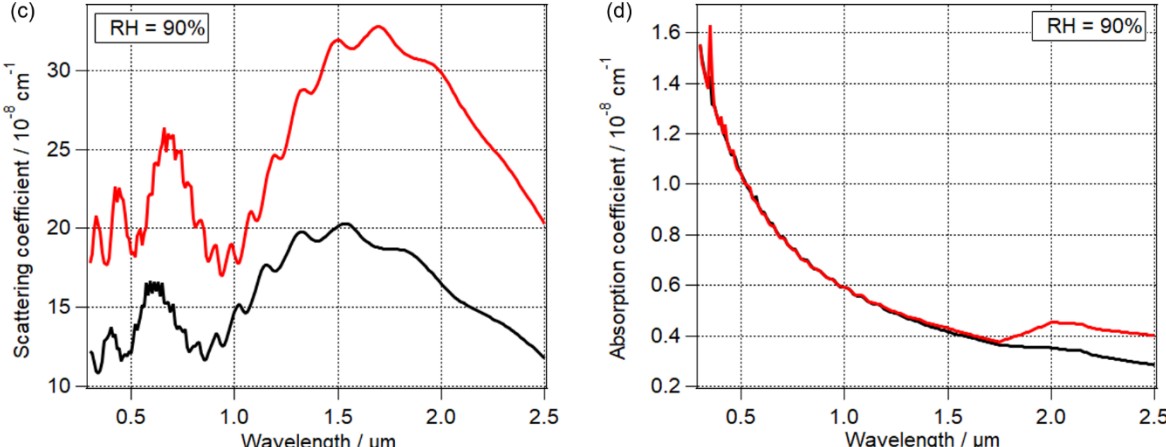

**Figure 4.** The effect of PBAP hygroscopicity (κ) on (a) scattering coefficient, (b) absorption coefficient of total particles at RH = 10% ($S_{opt7}$, $S_{opt8}$), (c) scattering coefficient, and (d) absorption coefficient of total particles at RH = 90% ($S_{opt9}$, $S_{opt10}$). The black lines indicate κ = 0.03 and the red lines indicate κ = 0.25 for all panels.

In addition to hygroscopicity ($κ_{PBAP}$), we explore the importance of biosurfactants which decrease surface tension of particles ($σ_{PBAP}$). A lower surface tension leads to a reduced particle curvature which, in turn, enhances the water uptake. Numerically, this is expressed in the Köhler equation:

$$s = \exp\left(\frac{A(σ)}{D_{wet}} - \frac{B(κ)}{D_{wet}^3}\right) \tag{2}$$

where $s$ is the equilibrium water vapor saturation ratio, $D_{wet}$ the wet particle diameter, the first term in the parentheses is the Kelvin (curvature) term which is a function of surface tension ($σ_{PBAP}$) following ***Equation 3*** and the second term is the Raoult (solute) term which can be parameterized by $κ_{PBAP}$ (Rose et al., 2008) following ***Equation 4***:

$$Kelvin\ term = \frac{A(σ)}{D_{wet}} = \frac{4σ_{sol}M_ω}{ρ_ω RTD_{wet}} \tag{3}$$

$$Raoult\ term = \frac{B(κ)}{D_{wet}^3} = -\ln\frac{D_{wet}^3 - D_s^3}{D_{wet}^3 - D_s^3(1-κ)} \tag{4}$$

where $σ_{sol}$ is surface tension of solution droplet (72 mN m$^{-1}$); $M_ω$ is molar mass of water (18 g mol$^{-1}$); $ρ_ω$ is density of water (1 g cm$^{-3}$); R is the universal gas constant (8.31 · 10$^7$ g cm$^2$ s$^{-2}$ K$^{-1}$ mol$^{-1}$); T is the absolute temperature (K); $D_{wet}$ is droplet diameter (cm); and $D_s$ is the diameter of the dry particle (cm).

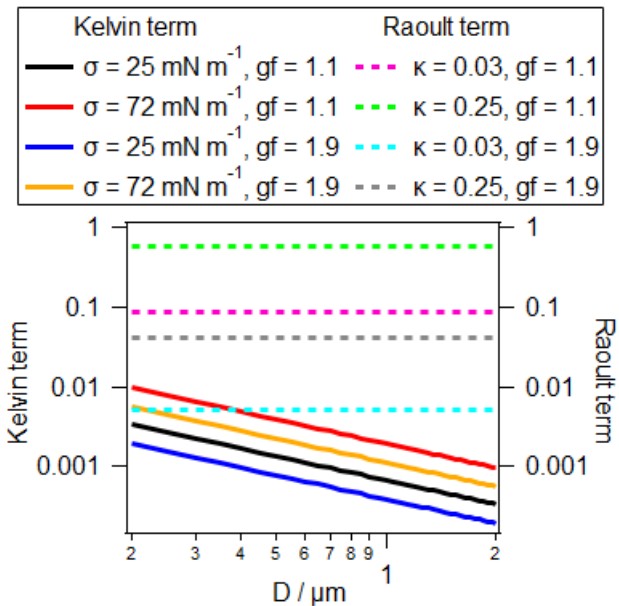

**Figure 5.** Kelvin term as a function of surface tension ($\sigma_{PBAP}$) for the $\sigma$ range as found for PBAPs (left axis; solid lines). Raoult term as a function of hygroscopicity ($\kappa_{PBAP}$) for the range of $\kappa$ as found for PBAPs (right axis; dashed lines).

The comparison of the two dimensionless terms shows that in most of the cases, the Raoult term exceeds the Kelvin term by at least one order of magnitude. Only for very small PBAPs, i.e. representative for viruses, SPPs or SFPs (***Section 2.1***), the curvature term significantly influences *s* (***Figure 5***). Based on this analysis, we can conclude that (bio)surfactants likely do not have a significant impact on the hygroscopic growth of PBAPs. A coating with surfactants might slow down the kinetics of the water uptake by particles (Davidovits et al., 2006). However, since the growth time scales of particles at RH < 100% are usually relatively long, the impact of surfactants on the time scale to reach equilibrium sizes is likely small, leading to a small importance of the effect of surfactant on water uptake and the corresponding optical properties.

### 4.1.3 Influence of complex refractive index ($m_{PBAP} = n + ik$) on scattering and absorption

The complex refractive index of PBAPs can be explained by their building blocks of various functional groups (Hill et al., 2015). Here the complex refractive indices of PBAPs are based on the measurements of *Erwinia herbicola* by Arakawa et al. (2003) and twelve other PBAPs by Hu et al. (2019); the complex refractive indices of 'other particles' in the model are the averaged values based on the volume fractions of ammonium sulfate, soot, and water (***Table 2***). The calculated scattering and absorption coefficients of the total particle population are shown in ***Figure 6***. Scattering coefficients for different PBAPs vary by a factor of up to four and the absorption coefficients by a factor of up to six.

The difference of optical properties between bacteria species or fungi species can be larger than that between these two types of PBAPs. Therefore, detailed information on PBAP species is important in order to estimate their direct interaction with radiation (**Section 4.1.4**).

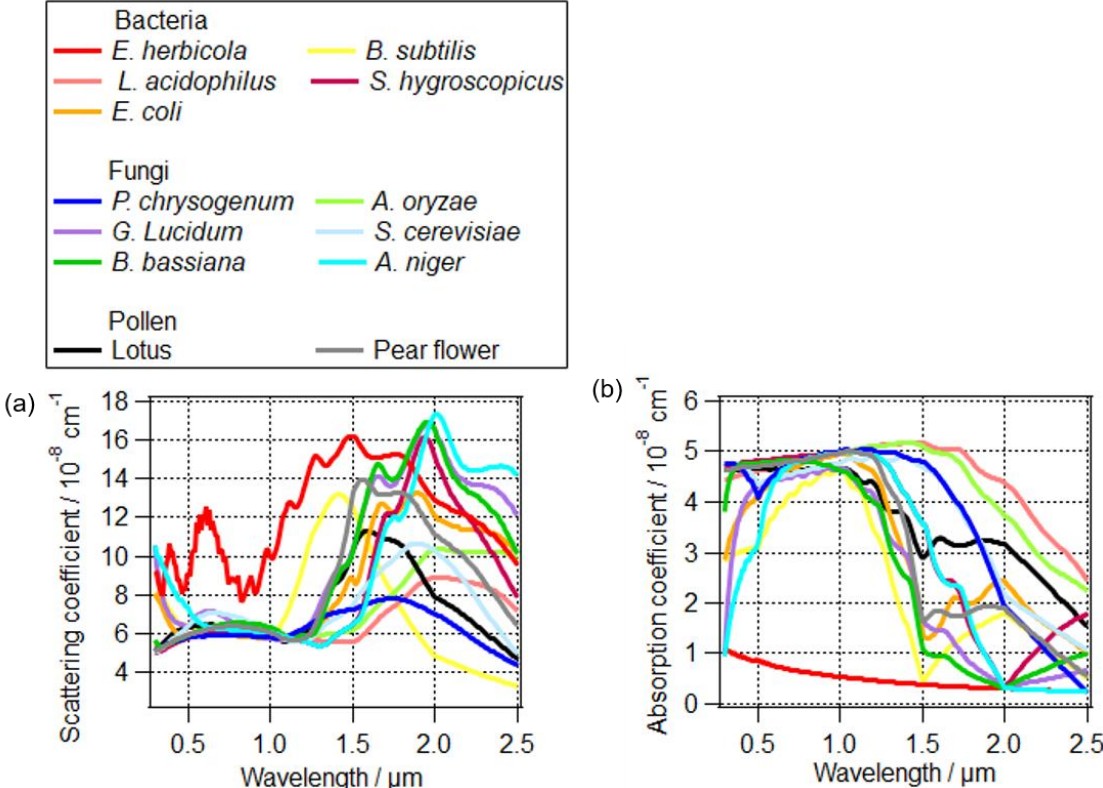

**Figure 6.** The influence of different PBAPs on (a) the scattering coefficient and (b) absorption coefficient of total particles. The refractive indices are based on the measurements by Arakawa et al. (2003) and Hu et al. (2019).

In addition to the variability in refractive index due to PBAP types, chemical processing of the molecules at the PBAP surface might modify the refractive index. It has been shown that nitration of SOA, i.e. the addition of a nitro group, leads to the formation of brown carbon (Moise et al., 2015). Qualitatively, it has been demonstrated that proteins can be nitrated, similar to SOA compounds (Shiraiwa et al., 2012). Due to the lack of data on the change of complex refractive index ($\Delta m$) for nitrated proteins in PBAPs, we assume PBAPs have a similar change in the refractive index to that of SOA ($S_{opt12}$ and $S_{opt13}$). After nitration, the scattering coefficient decreases by ~20% in the range of 300 nm $< \lambda <$ 450 nm and is nearly constant in the range of 460 nm $< \lambda <$ 560 nm (***Figure 7a***). The scattering coefficient depends non-linearly on the real and the imaginary parts. The absorption coefficient of nitrated PBAPs is higher by 14% to 160% in the range of 300 nm $< \lambda <$ 540 nm (***Figure 7b***) and is nearly constant in the range of 550 nm $< \lambda <$ 560 nm. The largest difference (~160%) for absorption coefficient is observed at 440 nm and the smallest difference (~6%) is

observed at 560 nm, which can be attributed to the wavelength-dependent change of the imaginary part ($\Delta k$) (Liu et al., 2015). The assumptions on $\Delta m$ made for the simulations shown in ***Figure 7*** are likely an overestimate of the chemical processing of PBAP constituents since (1) experimental conditions are often optimized so that a large fraction of particles is nitrated (Liu et al., 2015), as opposed to ~0.1% of nitrated proteins observed in the atmosphere (Franze et al., 2005), (2) we assume nitration to occur over the whole residence time of particles in the atmosphere while proteins can be nitrated only under conditions of sufficiently high $NO_x$ levels (Shiraiwa et al., 2012), and (3) a rather high concentration of $N_{PBAP} = 1$ cm$^{-3}$ is considered.

While generally, light-absorbing organics ('brown carbon') might contribute to the aerosol semi-direct effect (Brown et al., 2018; Hansen et al., 1997), i.e. the impact of aerosol heating on clouds, it seems unlikely that PBAPs have a significant contribution to it. Given the supermicron sizes of most PBAPs, their concentration decreases strongly as a function of altitude (Ziemba et al., 2016) and thus their concentration near cloud tops is likely negligible.

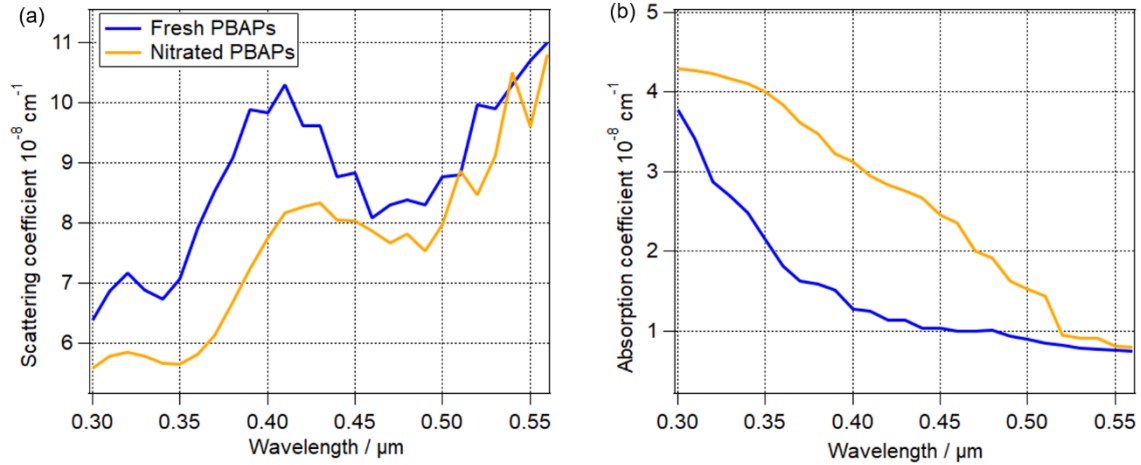

**Figure 7.** The influence of nitration on (a) the scattering coefficient and (b) absorption coefficient of total particles. The blue and brown lines indicate fresh PBAPs ($S_{opt12}$) and nitrated PBAPs ($S_{opt13}$), respectively.

### 4.1.4 Estimate of change of radiative forcing introduced by PBAPs

The direct radiative effect of particles can be expressed in terms of the single scattering albedo (SSA, i.e. the ratio of scattering coefficient to extinction coefficient) and radiative forcing efficiency (RFE, i.e. radiative forcing per unit optical depth) (Dinar et al., 2007; Randles et al., 2004). The RFE at 390 nm and 532 nm can be calculated as (Dinar et al., 2007):

$$RFE = S_{con}D_{len}(1 - A_{cld})T_{atm}^2(1 - R_{sfc})^2\left[2R_{sfc}\frac{1-\omega}{(1-R_{sfc})^2} - \beta\omega\right] \qquad (5)$$

where $S_{con}$ is the solar constant (1370 W m$^{-2}$); $D_{len}$ is the fractional day length (0.5); $A_{cld}$ is the fractional cloud cover (0.6); $T_{atm}$ is the solar atmospheric transmittance (0.76), and $R_{sfc}$ is surface albedo (0.15); $\omega$ is the single scattering albedo (SSA), which is the ratio of scattering coefficient to extinction coefficient; $\beta$ is average upscatter fraction, which can be calculated as:

$$\beta = 0.082 + 1.85b - 2.97b^2 \tag{6}$$

$$b = \frac{1-g^2}{2g}\left[\frac{1}{\sqrt{1+g^2}} - \frac{1}{1+g}\right] \tag{7}$$

where b is the ratio of backscattering to scattering coefficient, g is the asymmetry factor which is assumed as 0.65 as an average of ambient measurements (~0.59-0.72 (Andrews et al., 2006)). The calculated RFE are listed in *Table 3* for some of the simulations (input parameters are listed in *Table 2*). The first row is the reference with internally mixed ammonium sulfate/soot particles while PBAPs are absent. As expected,

when $\lambda$ increases from 390 nm to 532 nm, SSA increases due to less efficient absorption in the visible wavelength range (Kirchstetter et al., 2004).

*Table 3.* Radiative forcing efficiency (RFE) at 390 nm and 532 nm calculated based on *Equations 5, 6, and 7* (Dinar et al., 2007). Some typical conditions are shown here to demonstrate the influence of various PBAP properties such as concentration, size, and complex refractive index.

| Simulation | SSA | RFE (W m$^{-2}$) | ΔRFE (W m$^{-2}$) | SSA | RFE (W m$^{-2}$) | ΔRFE (W m$^{-2}$) |
|---|---|---|---|---|---|---|
| | 390 nm (ultraviolet) | | | 532 nm (visible) | | |
| $S_{opt1}$ (without PBAPs, reference) | 0.643 | -0.5 | - | 0.728 | -6.84 | - |
| $S_{opt2}$ (N = 0.01 cm$^{-3}$, D = 1 μm, $m_{E.\ herbicola}$) | 0.646 | -0.72 | -0.22 | 0.73 | -6.99 | -0.15 |
| $S_{opt3}$ (N = 0.1 cm$^{-3}$, D = 1 μm, $m_{E.\ herbicola}$) | 0.668 | -2.36 | -1.86 | 0.747 | -8.26 | -1.42 |
| $S_{opt5}$ (N = 0.1 cm$^{-3}$, D = 2 μm, $m_{E.\ herbicola}$) | 0.738 | -7.59 | -7.09 | 0.791 | -11.54 | -4.68 |
| $S_{opt8}$ (N = 1 cm$^{-3}$, D = 2 μm, $m_{E.\ herbicola}$) | 0.917 | -20.94 | -20.44 | 0.927 | -21.68 | -14.84 |
| $S_{opt11}$ (N = 1 cm$^{-3}$, D = 2 μm, $m_{B\ subtilis}$) | 0.539 | 7.26 | 7.76 | 0.56 | 5.7 | 12.54 |
| $S_{opt12}$ (N = 1 cm$^{-3}$, D = 2 μm, $m_{Fresh\ PBAP}$) | 0.868 | -17.29 | -16.79 | 0.927 | -21.69 | -14.85 |
| $S_{opt13}$ (N = 1 cm$^{-3}$, D = 2 μm, $m_{Nitrated\ PBAP}$) | 0.692 | -4.15 | -3.65 | 0.909 | -20.34 | -13.5 |

The RFE values in *Table 3* only represent radiative forcing of a small range of particle sizes and a constant composition and number concentration of other particles; however, the differences (ΔRFE) allow evaluating the relative importance of the various PBAP parameters ($N_{PBAP}$, $D_{PBAP}$, $m_{PBAP}$) in terms of their direct interaction with radiation. A negative ΔRFE implies more scattering and a positive ΔRFE implies more absorption due to the presence of PBAPs.

With a typical concentration of $N_{Erwinia\ herbicola} = 0.01$ cm$^{-3}$ ($S_{opt2}$), the SSA increases and the RFE is more negative by 44% and 2% at $\lambda = 390$ nm and $\lambda = 532$ nm, respectively, as compared to the reference case ($S_{opt1}$ without PBAPs). With a higher number concentration of $N_{Erwinia\ herbicola} = 0.1$ cm$^{-3}$ ($S_{opt3}$), the RFE becomes more negative by 228% and 18% at 390 nm and 532 nm, respectively, as compared to the low number concentration $N_{Erwinia\ herbicola} = 0.01$ cm$^{-3}$ ($S_{opt2}$). When the diameter increases to D = 2 µm ($S_{opt5}$), the

RFE is more negative by 221% and 40% at 390 nm and 532 nm, respectively, as compared to the D = 1 µm ($S_{opt3}$). The above results suggest that (1) both the concentration and the size of PBAPs can enhance the RFE significantly and (2) PBAPs affects the optical properties more at the UV wavelength of 390 nm than at the visible wavelength of 532 nm.

    All PBAPs for which refractive indices are listed in **Table 1**, show a wavelength dependence on scattering

and absorption. The imaginary part (k) varies by three orders of magnitude between different PBAPs (***Table 1***), which makes both the sign and the absolute value of the direct radiative effects of PBAPs uncertain. For example, both *Erwinia herbicola* and *Bacillus subtilis* have been found in the atmosphere (Després et al., 2012). *E. herbicola* is expected to induce more scattering ($S_{opt8}$) whereas *B. subtilis* is expected to induce more absorption ($S_{opt11}$). Due to the lack of data of nitrated PBAPs, we used the refractive index of nitrated

SOA and fresh SOA (Liu et al., 2015) to represent nitrated PBAPs and fresh PBAPs. Compared to the fresh PBAPs ($S_{opt12}$), the nitrated PBAPs ($S_{opt13}$) cause less change of RFE, which can be explained by the increase of k for nitrated PBAPs due to the formation of brown carbon.

    Note that in the above simulations relatively high concentrations of PBPAs were assumed and should only be used to compare the relative importance of PBAP size and complex refractive index for their optical

properties. The properties of PBAPs can vary depending on species of PBAPs and ageing processes. Given that the number concentration of PBAPs is generally small, the direct radiative effect of PBAPs is likely restricted to small spatial scales.

## 4.2 Sensitivity of CCN activity to PBAP properties

### 4.2.1 Influence of PBAP diameter (D$_{PBAP}$) on CCN activation

The critical saturation $s_c$ can be used as a measure to estimate whether a particle will be activated into a cloud droplet (Rose et al., 2008):

$$s_c = \exp\left(\sqrt{\frac{4A^3}{27\kappa D_S{}^3}}\right) \tag{8}$$

where A can be found in ***Equation 5***, κ is hygroscopicity, and D$_s$ (cm) is mass equivalent diameter of dry solute particle. Applying this equation, one finds that for particles with D$_{PBAP}$ of 0.01 to 10 µm, the critical

supersaturations ($S_c = (s_c\text{-}1) \cdot 100\%$) are in a broad range of 0.0007%-24% (assuming κ = 0.03; σ = 72mN

m$^{-1}$). For large PBAPs with $D_{PBAP} > 0.5$ µm, the critical supersaturations $S_c$ is smaller than 0.062%. Typical environmental supersaturations ($S_{env}$) in stratocumulus and convective cumulus clouds are in the range of ~0.1-0.5% and ~0.5-1%, respectively (Pruppacher and Klett, 1997). Comparison to $S_{c,PBAP}$ shows that most PBAPs ($D_{PBAP} > 0.5$ µm) are likely activated to droplets as their $S_c$ are significantly smaller than $S_{env}$ in clouds.

### 4.2.2   Influence of the hygroscopicity ($\kappa_{BAP}$) and surface tension ($\sigma_{BAP}$) on CCN activation

*Figure 8a* shows the range of critical supersaturation ($S_c$) for the κ values shown in *Table 2* for the small PBAPs with $D_{PBAP} = 500$ nm, 100 nm, and 50 nm. For $D_{PBAP} = 500$ nm, $S_c$ is 0.02% ($\kappa_{PBAP} = 0.25$, $S_{CCN1}$) or 0.06% ($\kappa_{PBAP} = 0.03$, $S_{CCN2}$), which are both below typical environmental supersaturation ($S_{env}$) in clouds. Only for smaller PBAPs such as viruses, SPPs or SFPs with $D_{PBAP} = 100$ nm ($S_{CCN3}$, $S_{CCN4}$), $S_c$ changes from 0.24% ($S_{CCN3}$) to 0.69% ($S_{CCN4}$) when $\kappa_{PBAP}$ increases from 0.03 to 0.25. For even smaller $D_{PBAP}$ (50 nm), $S_c$ increases from 0.68% ($S_{CCN5}$) to 1.97% ($S_{CCN6}$) when $\kappa_{PBAP}$ increases from 0.03 to 0.25. Thus, only for fairly small PBAPs such as viruses, SPPs or SFPs (D ≤ 100 nm), the hygroscopicity $\kappa_{PBAP}$ may impact their CCN activation. Steiner et al. (2015) reported critical supersaturations ($S_c$) of 0.81 (± 0.07)% for 50-nm SPPs and 0.26 (± 0.03)% for 100 nm SPPs. These values are similar to the values discussed above (0.68% to 1.79% for 50-nm particles, 0.24% to 0.69% for 100-nm particles) and are also in agreement with values based on the hygroscopicity (0.1 ≤ $\kappa_{SPP}$ ≤ 0.2) reported by Mikhailov et al. (2019, 2020).

Overlaid on the vertical lines for $S_c$ in *Figure 8a* are $S_{env}$ in the cloud as calculated in our parcel model for different updraft velocities (w = 10 cm s$^{-1}$, 100 cm s$^{-1}$, and 300 cm s$^{-1}$). The sensitivity of CCN properties to updraft velocity and $S_{env}$ has been discussed in numerous previous studies, e.g., Ervens et al. (2005). *Figure 8a* corroborates the conclusions from these previous studies that the variation of the κ over wide ranges only introduces a small change in the CCN activity and in cloud properties (e.g., drop number concentration, LWC) and that particle composition is most important in clouds with low updraft velocities.

Similar to $S_c$ ranges due to different $\kappa_{PBAP}$ values, we compare in *Figure 8b* predicted $S_c$ ranges due to different values of $\sigma_{PBAP}$ for high biosurfactant concentrations (when mass fraction of surfactants to total particle mass > 0.1%, $\sigma_{BAP} = 25$ mN m$^{-1}$) to those predicted for very low surfactant concentrations ($\sigma_{BAP} = 72$ mN m$^{-1}$). For PBAPs with $D_{PBAP} = 500$ nm, $S_c$ changes from 0.01% ($S_{CCN7}$, $\sigma_{PBAP} = 25$ mN m$^{-1}$) to 0.06% ($S_{CCN2}$, $\sigma_{PBAP} = 72$ mN m$^{-1}$). As discussed before, these large PBAPs will be likely all activated in clouds and the small difference in $S_c$ introduced by change of surface tension ($\Delta\sigma$) does not cause a difference in their CCN ability. For smaller PBAPs, such as viruses, SFPs and SPPs with $D_{PBAP} = 100$ nm, $S_c$ changes from 0.14% ($S_{CCN8}$) to 0.69% ($S_{CCN4}$) when $\Delta\sigma_{PBAP} = 47$ mN m$^{-1}$. When $D_{PBAP}$ further decreases to 50 nm, $S_c$ changes from 0.4% ($S_{CCN9}$) to 1.97% ($S_{CCN6}$) when $\Delta\sigma_{BAP} = 47$ mN m$^{-1}$. Therefore, the effect of biosurfactant needs to be considered for small PBAPs in terms of CCN activity if a sufficiently large mass fraction of

strongly surface-active biosurfactant is present. Note that the assumption of $\sigma_{PBAP} = 25$ mN m$^{-1}$ in ***Figure 8b***

likely represents an overestimate as most biosurfactants exhibit a range of 30 mN m$^{-1}$ < $\sigma_{BAP}$ < 55 mN m$^{-1}$ (Renard et al., 2016). In addition, the biosurfactant concentration, and thus the surface tension according to ***Equation 1***, depends on the mass fraction of biosurfactants in the PBAPs, the growth factor and on diameter of PBAPs. If the mass fraction is very low, $\sigma_{PBAP} = 72$ mN m$^{-1}$; when the mass fraction of biosurfactants approaches ~0.1%, $\sigma_{BAP}$ might be as low as 25 mN m$^{-1}$. Typical surfactant mass concentrations are on the order of ~0.1% (Gérard et al., 2019); mass fractions for specific biosurfactants have not been determined yet. Such low mass fraction implies that only a few (< 10 to 100) surfactant molecules (with a molecular weight M ~1000 g mol$^{-1}$) are present on submicron particles and/or that only a fraction of particles is completely covered by surfactants and thus exhibits a reduced surface tension. While biosurfactants might be also taken up by other particles while they reside on surfaces (soil, vegetation) where PBAPs are active, our conclusions also hold for such particles. Our sensitivity studies show relatively lower sensitivity of cloud properties to particle composition than that predicted based on equilibrium conditions, in agreement with previous sensitivity studies (Ervens et al., 2005). Therefore, previous estimates of surfactant effects on cloud properties that are based on a simplified assumption of equilibrium conditions in clouds (Facchini et al., 1999), led to an overestimate of the role of surfactants on CCN.

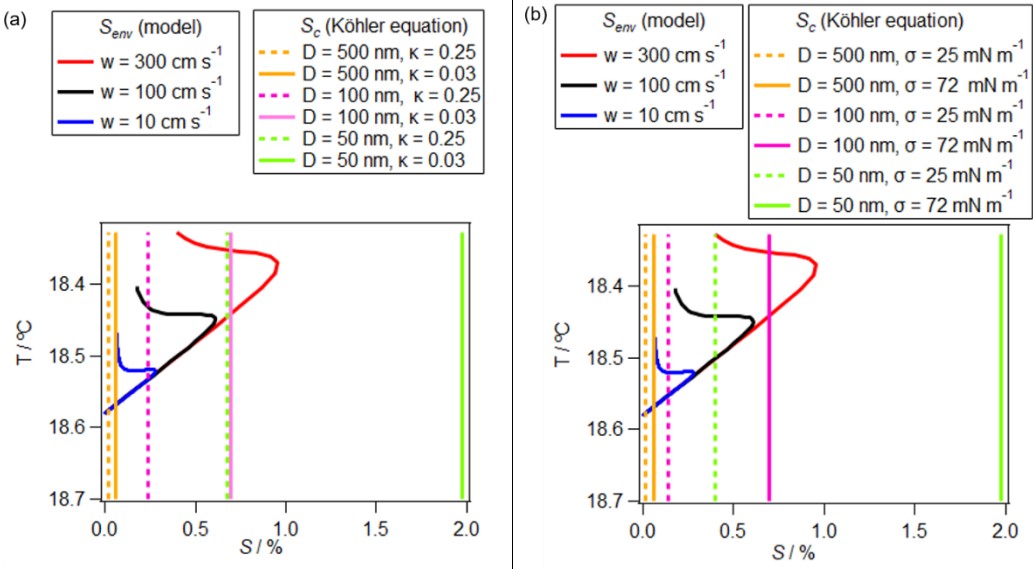

**Figure 8.** Comparison of the environmental supersaturation within the cloud ($S_{env}$) as predicted by the parcel model for different updraft velocities (w) to the critical supersaturation ($S_c$) of PBAPs based on Köhler theory. Results are shown as a function of (a) hygroscopicity parameters $\kappa_{PBAP}$ and (b) surface tension $\sigma_{PBAP}$. Input parameters to the parcel model are listed in ***Table 2***.

We conclude that the mass concentration of biosurfactants needs to be quantified in order to better explore the biosurfactant effect on CCN activation of small particles. Given that the surface concentration of

surfactants is likely higher than the bulk concentration (Bzdek et al., 2020; Lowe et al., 2019; Ruehl et al., 2016) as assumed here, even a smaller mass fraction of biosurfactants than calculated by *Equation 1* might be sufficient to decrease the surface tension of small aqueous PBAPs and the corresponding critical

supersaturation. However, also for the concept of surface partitioning of biosurfactants, rather than for a bulk concentration, our conclusions hold true on the limited impact of surface tension suppression on CCN activation of supermicron PBAPs.

### 4.3 Sensitivity of mixed-phase cloud evolution to PBAP properties

### 4.3.1 Influence of PBAP concentration ($N_{PBAP}$) and diameter ($D_{PBAP}$) on ice nucleation

$N_{PBAP}$ is on the same order of magnitude as that of total IN in some regions and at high temperatures (Pratt et al., 2009; Prenni et al., 2009), which makes PBAPs play an important role in mixed-phase clouds. Especially at the relatively high temperatures of T > -10 °C, some bacteria and fungi have much higher nucleation site density than other aerosol particles (Atkinson et al., 2013; Hoose and Möhler, 2012; Maters et al., 2019), and therefore $N_{PBAP, IN}$ / $N_{IN}$ is ~100%. *Figure 9a* shows the change of percentage contribution

of ice water content (% IWC, solid lines) and liquid water content (% LWC, dashed lines) to total adiabatic water content in a mixed-phase cloud ($S_{IN1}$, $S_{IN2}$, $S_{IN3}$). We define the onset of the Bergeron-Findeisen process as the temperature, at which the liquid water content fraction starts to efficiently decrease. With $N_{PBAP} = 1$ cm$^{-3}$, above an IWC contribution of ~3%, ice particles start growing at the expense of liquid water (Bergeron-Findeisen-Process) ($S_{IN1}$). At lower $N_{PBAP} = 0.01$ cm$^{-3}$ ($S_{IN2}$), the onset of the Bergeron-Findeisen-

Process starts slightly later. With $N_{PBAP} = 0.001$ cm$^{-3}$ ($S_{IN3}$), both IWC and LWC are predicted to increase simultaneously throughout the whole cloud, i.e. the Bergeron-Findeisen-Process is not initiated and cloud glaciation does not take place.

In *Figure 9b*, we compare model results for simulations $S_{IN4}$ and $S_{IN5}$ in order to explore the effect of $D_{PBAP}$. With larger PBAP size such as $D_{PBAP} = 2$ µm ($S_{IN4}$) or $D_{PBAP} = 5$ µm ($S_{IN5}$), ice formation starts earlier in the

cloud, but the onset of the Bergeron-Findeisen process occurs at approximately the same temperature as for smaller $D_{PBAP}$ because of the feedbacks of IWC and LWC on the supersaturation in the cloud and vice versa. For SPPs and SFPs with D ≤ 100 nm, immersion freezing may be limited by the droplet formation on these particles (*Figure S3*). As ice formation is less efficient on non-activated particles ('condensation freezing'), the onset temperatures of freezing is significantly lower. As supermicron particles likely act as CCN under

most conditions, this limitation might be smaller for large PBAPs.

It should be noted that our adiabatic parcel model framework cannot fully represent the complexity of all processes occuring in mixed-phase clouds, such as complete glaciation followed by precipitation and demise of the cloud. However, we rather demonstrate the relative changes in percentage contribution of ice water content (%IWC, solid lines) and liquid water content (%LWC, dashed lines) to total adiabatic water content

near the onset of ice nucleation. Thus, we apply our model in a similar way as in previous parcel model studies that explored the onset of the Bergeron-Findeisen process to various aspects of ice nucleation (Diehl et al., 2006; Eidhammer et al., 2009; Ervens et al., 2011; Khvorostyanov and Curry, 2005; Korolev, 2007; Korolev and Isaac, 2003).

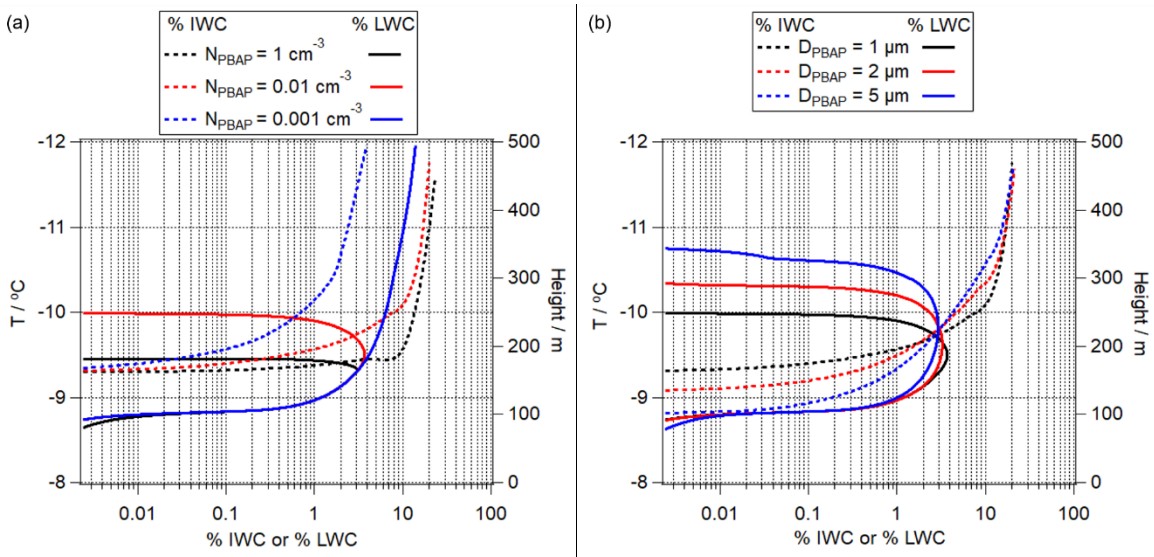


**Figure 9.** Percentage contribution of ice water content (%IWC, dashed lines) and liquid water content (%LWC, solid lines) to the total adiabatic water content as a function of (a) $N_{PBAP}$ and (b) $D_{PBAP}$. Details on the simulations can be found in ***Table 2***.

**4.3.2 Influence of the contact angle ($\theta_{BAP}$) on ice nucleation**

PBAPs exhibit a wide range of contact angles of $4° < \theta_{PBAP} < 44°$ (***Table 1***). ***Figure 10*** compares the predicted relative contributions of %IWC and %LWC to the total adiabatic water content. The comparison of Figures 10a and 10b shows that the onset temperatures of the %LWC decrease are at ~ -7.7 °C ($\theta_{PBAP} = 4°$) and ~ -8.3 °C ($\theta_{PBAP} = 20°$), respectively, i.e. resulting in a difference of $\Delta T$ ~0.6 °C. This difference is
predicted to be larger ($\Delta T$ ~3.3 °C) for PBAPs with $\theta_{PBAP} = 40°$.

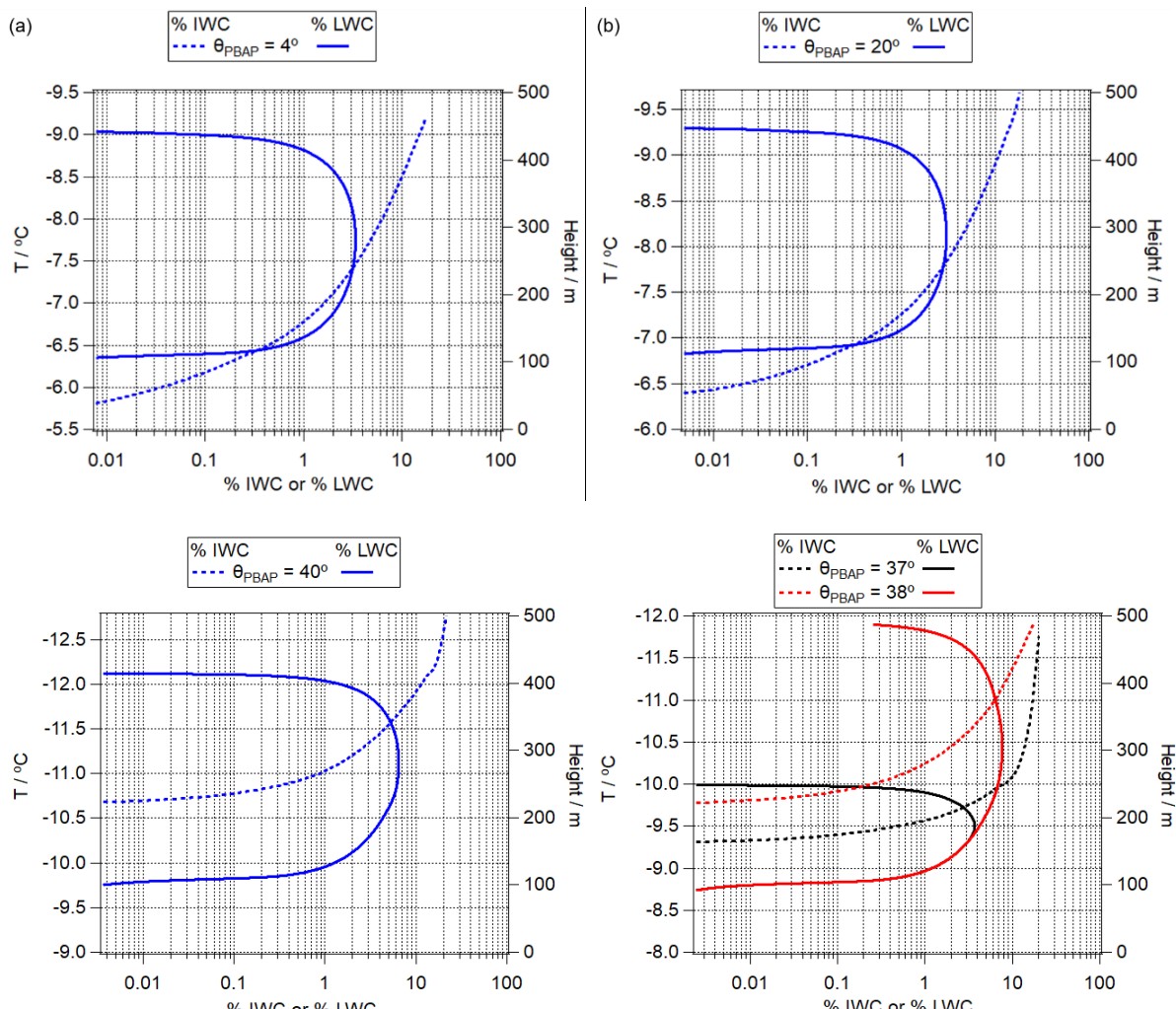

**Figure 10.** Percentage contribution of ice water content (%IWC, dashed lines) and liquid water content (%LWC, solid lines) to total adiabatic water content for $\theta_{PBAP}$ of (a) 4°; (b) 20°; (c) 40° and (d) 37° and 38°. The curves in the first three panels exhibit similar shapes for different temperature ranges, i.e. the Bergeron-Findeisen process starts at different temperatures. The last panel shows that even when the contact angle increases by 1°, the temperature, at which the %LWC fraction starts decreasing, differs significantly.

As discussed in *Section 2*, chemical (e.g., nitration, oxidation, adjustments due to pH) or physical processing of IN surfaces might lead to $\Delta\theta_{PBAP} \sim 1°$. In *Figure 10d*, we show %IWC and %LWC by comparing $S_{IN2}$ and $S_{IN9}$. The results show that even such a small change of 1° in $\theta$ can cause a significant difference in the predicted IWC and LWC evolutions. The temperature, at which the %LWC starts decreasing differs by $\Delta T$ ~1.3 °C. Such a change in $\theta$ may be induced by pH changes; for example, it was found that $\Delta\theta$ is ~1.5° for bacteria such as *Pseudomonas syringae* when the cells were exposed to solutions of pH 7.0 and 4.1 at temperatures of T > -10 °C. Denaturation of IN protein's agglomerates (polymers) occurs at pH below 4.5

(Schmid et al., 1997; Turner et al., 1990), suggesting that changes in IN activities due to pH might be reversible at least above this pH value.

Similar differences in θ could be also caused due to other processes, such as the oxidation of pollen that lead to $\Delta\theta$ ~1.5° at T ~ -39 °C (Gute and Abbatt, 2018). However, at this much lower temperature, the sensitivity of the frozen fraction to $\Delta\theta$ decreases (Ervens and Feingold, 2013). Overall, our model results suggest that
a small change in the contact angle due to different types of PBAPs or due to ageing processes might have a large impact on ice nucleation in clouds. These differences might translate into feedbacks on other subgrid and dynamical processes in the cloud that amplify or reduce the efficiency of glaciation. However, such processes cannot be further explored in the adiabatic parcel model framework.

**5. Conclusions**

Based on our model sensitivity studies, we can rank the relative importance of the PBAP properties and processes in *Figure 1* for their aerosol-cloud interactions and optical properties. Given the limitations of our process models in terms of scales, dimensions and parameter spaces, our results should be considered as qualitative, rather than quantitative estimates; the focus of our study is the comparison of relative changes due to various physicochemical parameters. Several findings of our model sensitivity results repeat those
that have been drawn previously for other atmospheric particle types (Hoose and Möhler, 2012; McFiggans et al., 2005; Moise et al., 2015). However, in addition, unlike other atmospheric particles, PBAPs may constitute living microorganisms; thus, their properties may not only be modified by chemical and physical processes (marked in green and blue, respectively, in *Figure 11*), but also by biological processes (marked in red in *Figure 11*). To date, the extent to which these biological processes affect PBAP properties in the
atmosphere is not known due to the lack of suitable data sets for atmospheric models. Our sensitivity studies, in combination with *Figure 11*, give a first idea on which biological processes could modify relevant PBAP properties.

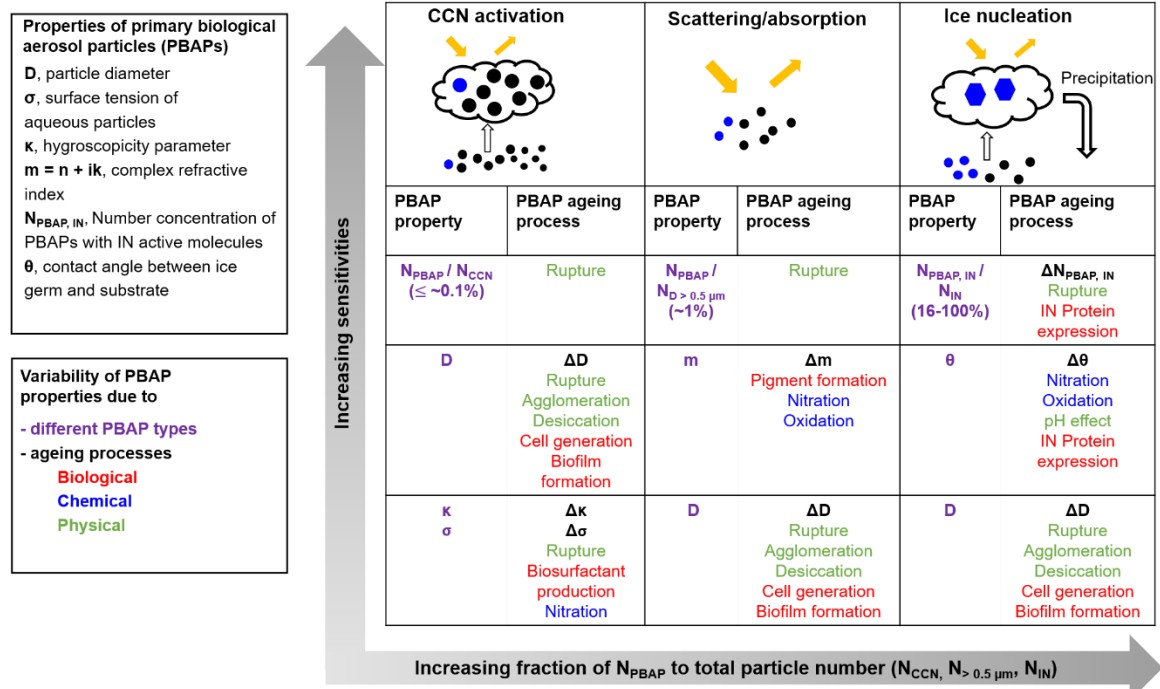

**Figure 11**. Schematic of PBAP types and ageing processes that affect their aerosol-cloud interactions and optical properties. The bottom arrow shows the increasing fraction of $N_{PBAP}$ to total particles ($N_{CCN}$, $N_{>5\mu m}$, and $N_{IN}$, respectively)**.** The left arrow indicates the increasing sensitivity to PBAP properties as predicted based on our process model studies. The various properties might be modified by physical (green), chemical (blue) and biological (red) ageing processes.

(1) For any climate-related effect, the number concentration of PBAPs ($N_{PBAP}$) is the most important parameter. The PBAP number concentrations assumed in our estimates are based on measurements near the ground (Huffman et al., 2012; Jaenicke, 2005; Tong and Lighthart, 2000; Whitehead et al., 2016), which typically decrease with altitude (Gabey et al., 2013; Perring et al., 2015; Ziemba et al., 2016). Thus, processes that affect $N_{PBAP}$ in the atmosphere need to be well constrained; these processes include not only direct emissions but also particle fragmentation (rupture) or possibly new cell generation (multiplication). The number fraction of PBAPs to total CCN is relatively small ($\leq$ ~0.1%). For example, in the Amazon, it is on the order of 0.01 to 0.1% based on the reported ranges of PBAP number concentrations ($0.2 < N_{PBAP} < 1.2$ cm$^{-3}$ (Whitehead et al., 2016); $0.04 < N_{PBAP} < 0.13$ cm$^{-3}$ (Huffman et al., 2012)) and CCN concentration ($N_{CCN}$ ~260 cm$^{-3}$, at 1% supersaturation (Roberts et al., 2001)). A similar ratio of $N_{PBAP}/N_{CCN}$ (~0.01 to 0.1%) can be derived based on measurements in the megacity Beijing with $N_{PBAP} \leq 1.4$ cm$^{-3}$ (Wei et al., 2016) during haze days and $N_{CCN} \leq 9.9 \cdot 10^3$ cm$^{-3}$ (at 0.86% supersaturation) (Gunthe et al., 2011). Thus, a small change in $N_{PBAP}$ likely does not significantly affect cloud droplet number concentration. Only in rare events, e.g. when pollen grains rupture with high efficiency, $N_{pollen}$ might considerably affect $N_{CCN}$ (Wozniak et al.,

2018). However, droplet formation on PBAPs increases microorganisms' survival rate and decreases their atmospheric residence time due to precipitation, so the knowledge of their CCN-relevant properties is of biological relevance.

PBAPs contribute ~1% to large particles with D > 0.5 µm (Zhang et al., 2019), which makes them relatively important for scattering/absorption at a limited range of wavelengths. Only in the presence of high $N_{PBAP}$, it is expected that they have (local) impacts on the direct aerosol effect.

The number concentration of PBAPs that nucleate ice at T > -10°C is on the order of $10^{-5}$ to $10^{-3}$ cm$^{-3}$ (Murray et al., 2012). PBAPs comprise the predominant fraction of atmospheric particles that efficiently nucleate ice at these temperatures, i.e. $N_{PBAP}/N_{IN}$ ~100% at T > -10°C (Hoose and Möhler, 2012). This fraction decreases at temperatures at which more abundant particles (such as dust) are also efficient ice nuclei: For example, at -30 °C, PBAPs contribute 16% to 76% (Prenni et al., 2009) or 33% (Pratt et al., 2009) to total IN in mixed-phase clouds. Lab measurements have shown that up to 100% of pollen grains have IN nucleating macromolecules on their surface, whereas only 0.01 to 10% of bacteria express the proteins or other macromolecules that initiate ice nucleation (Failor et al., 2017; Joly et al., 2013; Pummer et al., 2015).

(2) The size of PBAPs influences the effects in *Figure 11* to different extents: While it is likely the most important parameter to determine their ability to act as CCN compared to hygroscopicity and surface tension, its role for PBAPs' optical properties is smaller than that of the refractive index. Also PBAP size plays a less important role than surface properties in the efficiency of ice nucleation. While several biological processes may increase the size of PBAP (e.g. agglomeration, cell generation), these changes are likely not important for the CCN activity of supermicron PBAPs since they will be activated under most conditions and thus an increase in their size does not affect their CCN behavior. However, modifications in the size, hygroscopicity ($\kappa_{PBAP}$), and surface tension ($\sigma_{PBAP}$) of smaller PBAPs, such as viruses, SPPs and SPFs, can influence their CCN activation. $\kappa_{PBAP}$ might be modified by physical (e.g., release of inner molecules due to rupture of pollen and fungal spores, condensation of gases), biological (e.g., formation of biosurfactants or other metabolic products), and chemical (e.g., nitration, oxidation) processes. Thus, processes that modify hygroscopic or surface tension properties of these smaller PBAPs might significantly change their ability to take up water vapor and form cloud droplets.

(3) The optical properties of PBAP are mostly determined by their complex refractive index (m = n + ik), especially by the imaginary part (k) which varies by three orders of magnitude among PBAPs. Under conditions when PBAPs significantly affect Mie scattering, small variabilities in the refractive index due to PBAP types or ageing processes might enhance (or diminish) their direct interaction with radiation (scattering/absorption). Modification processes include pigment formation as a defense mechanism of bacteria to oxidative stress (Fong et al., 2001; Noctor et al., 2015; Pšenčík et al., 2004; Wirgot et al., 2017)

and nitration/oxidation of surface molecules (He et al., 2018; Liu et al., 2015; Nakayama et al., 2018). Additional biological processes such as biofilm formation are also included in *Figure 11* although experimental data are lacking to estimate their impact on PBAP optical properties.

(4) The ice nucleation activity of aerosol particles is often parameterized with a single contact angle ($\theta$) between the particle surface and ice. *Table 1* shows that $\theta$ significantly differs among different PBAP types. In addition, our model sensitivity studies suggest that even a small change ($\Delta\theta_{PBAP}$ ~1°) as caused by chemical processing of surfaces, pH change of the surrounding aqueous phase, or biological processes such as protein expression level might significantly affect this activity. At temperatures at which PBAPs are the predominant IN (T > -10 °C), such a small change might translate into large changes in the onset temperature of freezing and cloud glaciation can be affected. Thus, in order to comprehensively account for ice nucleation of PBAPs, not only various PBAP types, but also $\Delta\theta_{PBAP}$ due to modification by chemical and possibly biological processes should be considered in models.

Exceeding numerous recent review articles that highlight the importance of PBAPs in general (Coluzza et al., 2017; Després et al., 2012; Fröhlich-Nowoisky et al., 2016; Haddrell and Thomas, 2017; Šantl-Temkiv et al., 2020; Smets et al., 2016), *Figure 11* gives more specific guidance on future measurements of the most sensitive PBAP properties in terms of their interaction with radiation and with water vapor. The detailed knowledge of PBAP properties might be of limited importance for global radiative forcing estimates, but is also relevant to properly describe PBAP transport, dispersion and lifetime in the atmosphere, which eventually affects biodiversity (Morris et al., 2014) and public health (Fröhlich-Nowoisky et al., 2016). While previous studies only focused on the physical and chemical properties, we highlight the uniqueness of PBAPs undergoing biological processes to adapt to the harsh atmospheric conditions; such processes might affect the adaption of PBAPs to atmospheric conditions which impacts their survival, transport and dispersion in the atmosphere.

**Code and data availability:** Details on the model codes and further model results can be obtained from the corresponding author upon request.

**Author contributions:** MZ and BE designed the model framework. AK, PA, AD contributed by fruitful discussions and commented on the manuscript.

**Competing interests:** The authors declare that they do not have any competing interests.

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
