# Peer review of "Sensitivities to biological aerosol particle properties and ageing processes: Potential implications for aerosol-cloud interactions and optical properties"

_Atmospheric Chemistry and Physics, 2020_

## Referee Comment (RC1) · Anonymous Referee #1 · 28 Oct 2020

General comments: Overall, this paper is a useful study that investigates the relevant optical properties of biological aerosol particles. They provide some excellent comparison tests of which parameters and processes are important, and provide a framework for understanding these findings.

Major comments:

1. The authors frequently talk about how they do not intend this paper to be a comprehensive literature review (e.g., lines 137-140), yet it is still important that they cover the range of values that are found in the literature. Specifically, I would like to see an inclusion of more up to date information on pollen and fungal spore rupture (see next

comment)

2. The authors do not provide equal weight to the physical ageing via rupture of biological particles such as fungal spores and pollen. Physical ageing processes are noted, but they have not done the appropriate literature review to accurately capture how some types of biological particles may change. This represents an important atmospheric secondary process that can change both the size distribution as well as potentially the optical properties. This should be mentioned in the introduction when discussing "physical transformations" around line 100, and more specifically throughout the paper, particularly for including observed size distributions and their influence the optical properties. Pollen rupture is mentioned briefly on lines 168-169 and as a single referenced line item in Table 1, but this underestimates this process based on the long list of epidemiological literature on this process (e.g., Suphiolglu et al. 1992; Grote et al., 2001; Taylor et al. 2002; Taylor et al. 2004). More recently, fungal spores have been shown to rupture as well (Lawler et al., 2020; China et al., 2017), and this has not been mentioned at all in the text nor in Table 1. Overall, the authors spend a lot of time on the chemical processing (e.g., nitration) and its impacts, but very little on this physical process.

3. Overall, the sensitivity studies described are useful, but there was little discussion of box model results. Specifically, more detail on the following would enhance the paper: a. lines 367-369 – why does the absorption coefficient increase at the higher wavelengths? b. Figure 6 – large changes with refractive indices (no surprise) but hardly any discussion in the text of what changes are important c. Figure 7: why are the nitrated changes in scattering large at smaller wavelengths?

4. The ranking in Figure 11 is potentially useful but ultimately confusing. Please revise the accompanying text to make this figure more clear – right now the discussion is scattered and it would help to clarify this figure more, as it is ultimately very useful.

Minor comments: 1. The acronym used in the paper is inconsistent with the literature

on *primary* biological aerosol particles (PBAP) not BAP. While they do talk about some secondary processing of the aerosols, the origin of the particles is still primary (as opposed to secondary formation), and consistency with prior work is helpful.

2. Line 57 – is the Londahl et al. 2014 the correct reference here? This seems to be an error.

3. Line 58 – Myhre et al. 2013 is not in the reference list.

4. Line 98 – Pollen can also nucleate ice – see Diehl et al. 2001

5. Line 181 – missing word between "that" and "might"? Can't tell what this sentence is supposed to say.

6. Table 2: missing many references on the rupture of pollen. I actually think that these numbers are incorrect and very much mis-represent the range of potential sizes (see refs Grote, Taylor, Suphioulglu for a few; listed below). Also, you are missing the rupture of fungal spores (China, Lawler; see references below). Also missing the fact that the hygroscopicity of pollen may change on rupture (not just from oxidation).

7. Lines 283-284: The text should more clearly state that certain classes of PBAP are excluded based on the 0.5-2.8 micron size representation.

8. Line 342: Fungal fragments could also be on the order of this size. . .

9. Lines 360-363: This line downplays the potential importance of non-spherical particles. The true atmospheric range of moisture conditions is not enough to say what is more likely, therefore this speculation should be removed and it would be better to discuss what types of uncertainties non-spherical particles would include.

10. Figure 3: the caption states that there is an a/b panel to capture scattering and absorption, yet only the scattering is shown.

11. Line 392: "very small PBAP" could also be pollen or fungal fragments. Please see literature suggestions in the major comments.

12. Line 422: I think this is supposed to be S15 and S16?

13. Line 471: what is delta_mBAP? First use, please define. (perhaps including S13, S15 and S16?)

14. Table 3: last row – is dm_aged the same as dm_nitrated? Different terminology than Table 2.

15. Also in Table 3 – what is the dm actually referring to? Hard to tell from comparing with Table 2.

16. Lines 495-298: Could be compared with the observed values of Sc from Steiner et al. 2015

17. Line 500 – The missed rupture literature could also be important here. Physical processes like rupture could create many more hygroscopic particles.

18. Lines 568-571 – overall, these changes are really hard to see in the figure. Is it possible to overlay Figures 9a/b so they can be more directly compared?

19. Lines 630-632: What is the reference for this sentence "However, as it has been shown that at many locations NBAP/Ntotal is approximately constant…." – this is not true for fungal spores and pollen. The emissions of these types of PBAP are very spatially and temporally heterogeneous, and tend to be more event-based than consistent.

References China et al. (2016) Rupturing of biological spores as a source of secondary particles in the atmosphere, ES&T, 50, 22, 12179-12186.

Diehl, K., Quick, C., Matthias-Maser, S., Mitra, S. K., & Jaenicke, R. (2001). The ice nucleating ability of pollen part I: Laboratory studies in deposition and condensation freezing modes. Atmospheric Research, 58(2), 75–87. https://doi.org/10.1016/S0169-8095(01)00091-6

Grote, M., Vrtala, S., Niederberger, V., Wiermann, R., Valenta, R., & Reichelt, R. (2001). Release of allergen-bearing cytoplasm from hydrated pollen: A

mechanism common to a variety of grass (poaceae) species revealed by electron microscopy. Journal of Allergy and Clinical Immunology, 108(1), 109–115. https://doi.org/10.1067/mai.2001.116431

Lawler, M.J. et al. (2019) Atmospheric fungal nanoparticle bursts, Science Advances, 6,3, doi: 10.1126/sciadv.aax9051.

Steiner, A. L., Brooks, S. D., Deng, C., Thornton, D. C. O., Pendleton,M. W., & Bryant, V. (2015). Pollen as atmospheric cloud condensation nuclei. Geophysical Research Letters, 42, 3596–3602. https://doi.org/10.1002/2015GL064060

Suphioglu, C., Singh, M. B., Taylor, P., Knox, R. B., Bellomo, R., Holmes, P., & Puy, R. (1992). Mechanism of grass-pollen-induced asthma. The Lancet, 339(8793), 569–572. https://doi.org/10.1016/0140-6736(92)90864-Y

Taylor, P. E., Flagan, R. C., Miguel, A. G., Valenta, R., & Glovsky, M. M. (2004). Birch pollen rupture and the release of aerosols of respirable allergens. Clinical and Experimental Allergy, 34(10), 1591–1596. https://doi.org/10.1111/j.1365-2222.2004.02078.x

Taylor, P. E., Flagan, R. C., Valenta, R., & Glovsky, M. M. (2002). Release of allergens as respirable aerosols: A link between grass pollen and asthma. Journal of Allergy and Clinical Immunology, 109(1), 51–56. https://doi.org/10.1067/mai.2002.120759

---

## Referee Comment (RC2) · Anonymous Referee #2 · 19 Nov 2020

This work explores the radiative effects of biological aerosol particles (BAPs). The authors conduct a literature review on the physicochemical properties associated with scattering and absorption of radiation, cloud condensation nuclei (CCN) activation and ice nucleation efficiency of BAPs. From this they establish plausible ranges for different BAPs properties, then perform several sensitivity studies to roughly assess the possible impacts on radiation and cloud evolution, hence on climate. This is a well written paper that lies within the scope of ACP. However, it is also very speculative. It is not clear that enough data has been reported to establish a possible impact. The sensitivity studies are also conducted in a very idealized and simplified way, particularly those related to cloud formation. Thus, some clarification of the approach taken, as well as

the considerable limitations of this study, are required before it can be accepted for publication.

General Comments:

The authors carry out a somehow extensive literature review to select a plausible range of parameters to carry out sensitivity studies. This is commendable; however, the process studies sprung from it may be too limited and too idealized to be meaningful regarding the effect of BAPs on radiative forcing. For example, a parcel model is not at all appropriate to make conclusions on the prevalence of the Bergeron-Findeinsen (BF) process. This is further discussed below. The process studies thus lead to somehow obvious conclusions, which are not necessarily unique to BAPs and that are already well-known, i.e., higher kappa value leads to easier CCN activation, lower contact angle to more efficient ice nucleation, in a parcel model ice grows at the expense of liquid, higher refractive index leads to enhance absorption, and so on. Thus the authors must modify the language with a honest and thorough assessment of the limitations of their study and also emphasize differences/similarities with the typical behavior of other aerosols.

Detailed comments:

Line 11. Are biological fragments considered here?

Line 37. Delete "the"

Line 41. Maybe "in the urban area of Mainz" is more appropriate.

Line 68. In Figure 1 would absorption of solar radiation lead to a semi-direct effect?

Line 165. Must be "agglomerates"

Line 240. It is not clear what the maximum frozen fraction means here. If the temperature is lowered to -40 C the bacteria won't freeze at all? Why is -10 C the temperature of choice?

ACPD
Line 244. This is factually wrong. All real materials show stochastic behavior during ice nucleation. Please rephrase.

Line 250. The application of the contact angle approach to ice nucleation in biological materials is fraught with problems, since all the assumptions of classical nucleation theory break, and depends strongly on the values selected for other very uncertain parameters like for example the ice-liquid interfacial tension and the activation energy. Please add an explanation on the limitations of describing ice nucleation in biological materials.

Line 258. INAS is obtained by fitting freezing experiments neglecting the time dependency of ice nucleation. Please rephrase. I would suggest the authors refrain from discussing deterministic vs stochastic behavior since it is distracting and not at all clear what they mean, particularly for BAPs.

Line 273. Are these changes due to denaturation or are they reversible?

Line 286. Is there a reason to consider BAPs externally-mixed and monodisperse?

Line 301. What are the properties of the "other" aerosol. Is there any sensitivity of the results to this assumption?

Line 311. If the BAPs freeze by immersion, shouldn't they be inside the droplets? Are the results sensitive to Nother?

Line 313. This is a crude approximation that only works to make an assessment on droplet/ice formation, but would be very misleading to estimate LWC and IWC. Once ice is formed a whole set of other microphysical processes rapidly take place. Please justify why this approach is used at all.

Line 330. This "Nother" is different from the "Nother" of line 309, which is also different to the one in line 301.

Line 438. All of these values change strongly with location, so it is not clear why this
estimate is not given with a range of uncertainty, down in line 450.

Line 459. This would only be true if BAPs were uniform in the globe and isolated from other aerosols.

Line 479. There is certainly not data to support this "independence" assertion. The authors could probably make this assumption but clarify that it is in the absence of better data.

Line 509. I am not sure what is shown here. This caption needs more information, Table 2 does not even say what Senv or Sc are.

Line 531. I don't think this is a buffer effect, or at least explain what that means in this context.

Line 561. This conclusion is short-sighted. D influences droplet activation hence where freezing could occur. Mixed-phase clouds are CCN limited as well, so the effect may not be negligible.

Line 570. Please explain how the authors pin the BF effect to a particular Delta\_T (also what Delta\_T means). Is the T shift related to a later onset of freezing?

Line 574. No, this is not clear at all. Early freezing may result in early scavenging of available BAP and actually limiting instead of enhancing BF processes. There is a myriad of other things that can negate the onset of BF process, none of which can be represented in a parcel model: high subgrid scale vertical velocity, the presence of other efficient ice nucleating particles (for example feldspards can freeze at very high T as well), preferential spatial concentration of liquid and ice particles, to name a few. I would accept a much more cautious language like for example, "has the potential to affect the BF process" followed by a list of all the things that need to be addressed before this conclusion can be asserted with any degree of accuracy.

Line 584. Please show here where the BF process is initiated.
Line 596. Figure 11 must be removed. To start it is confusing since clearly the different aging processes affect more than one variable at a time. More fundamentally it presents a misleading, "final" assessment of something that is highly uncertain. The data is still too scarce and the studies way too idealized to support this figure.

Line 615. What about the semi-direct effect?

Line 630. See comment on Line 479.

Line 636. See comment on Line 240.

Line 654. This is speculation, since the authors do not perform any studies on cell generation.

---

## Author Comment (AC1) · 22 Dec 2020

*General comments: Overall, this paper is a useful study that investigates the relevant optical properties of biological aerosol particles. They provide some excellent comparison tests of which parameters and processes are important, and provide a framework for understanding these findings.*

*Major comments:*

*Referee Comment 1:*

*The authors frequently talk about how they do not intend this paper to be a comprehensive literature review (e.g., lines 137-140), yet it is still important that they cover the range of values that are found in the literature. Specifically, I would like to see an inclusion of more up to date information on pollen and fungal spore rupture (see next comment)*

**Responses and revisions 1:**

We thank the referee for this suggestion and agree that some discussion on pollen and fungal spore rupture should be included as an additional process. We added information on the ranges of sizes of fragments of pollen and fungal spores to the revised manuscript and indicated these changes in the following comments below.

*Comment 2:*

*The authors do not provide equal weight to the physical ageing via rupture of biological particles such as fungal spores and pollen. Physical ageing processes are noted, but they have not done the appropriate literature review to accurately capture how some types of biological particles may change. This represents an important atmospheric secondary process that can change both the size distribution as well as potentially the optical properties. This should be mentioned in the introduction when discussing "physical transformations" around line 100, and more specifically throughout the paper, particularly for including observed size distributions and their influence the optical properties. Pollen rupture is mentioned briefly on lines 168-169 and as a single referenced line item in Table 1, but this underestimates this process based on the long list of epidemiological literature on this process (e.g., Suphiolglu et al. 1992; Grote et al., 2001; Taylor et al. 2002; Taylor et al. 2004). More recently, fungal spores have been shown to rupture as well (Lawler et al., 2020; China et al., 2017), and this has not been mentioned at all in the text nor in Table 1. Overall, the authors spend a lot of time on the chemical processing (e.g., nitration) and its impacts, but very little on this physical process.*

**Responses and revisions 2:**

We thank the referee for pointing us to these references on the rupture of pollen and fungal spores as it contributes to ageing processes of pollen and fugal spores.

We add the following text:

- in the introduction at line 78:

In particular pollen rupture leads to a huge increase in the number of subpollen particles (SPPs) (Bacsi et al., 2006; Suphioglu et al., 1992; Taylor et al., 2004; Wozniak et al., 2018). By assuming that one pollen grain releases up to 106 SPPs, regional model studies suggested that the resulting SPPs can significantly suppress seasonal precipitation (Wozniak et al., 2018).

- at the end of 'Physical transformations' at line 105:

For example, the break-up of pollen or fungi due to rupture can lead to higher number concentrations by several orders of magnitude (Suphioglu et al., 1992; Wozniak et al., 2018).

- At the end of Section 2.1, we modified the text as follows at line 174:

At high RH and during precipitation or thunderstorms, pollen absorb water and one pollen grain can release $\sim 10^3$ SPPs due to osmotic pressure (Grote et al., 2001; Suphioglu et al., 1992).  This process can result in fragments with diameters of 1-4 μm and number concentrations of $N_{SPP}$ ~0.1 cm$^{-3}$ during thunderstorms (Zhang et al., 2019). These

concentrations correspond to ~1 to 25 ng m$^{-3}$ ($D_{SPP}$ < 2 μm) (Miguel et al., 2006). Laboratory chamber measurements have shown that SPPs from rupture of fresh birch pollen or grass pollen have diameters of in the range of 0.03 to 4.7 μm (Taylor et al., 2002, 2004). Recent laboratory measurements suggest that also fungal spores can rupture, resulting in subfungi particles (SFPs) with $D_{SFP}$ of 0.03 to 0.9 μm after exposure to high relative humidity (China et al., 2016). Ambient measurements suggest $N_{SFP}$ of 150 to 455 cm$^{-3}$ (10 nm < $D_{SFP}$ < 100 nm) after rainfall; observed peaks in aerosol size distributions at 20 nm < $D_{SFP}$ < 50 nm which frequently appeared 1.5 days after rain events were ascribed to such rupture events (Lawler et al., 2020).

- We modified the following sentences at the end of Section 2.3.1 at line 225:

The hygroscopity of pollen is similar to that of bacteria: The κ value of intact pollen grains falls into the range of $0.03 \leq \kappa_{pollen} \leq 0.17$ (Chen et al., 2019; Pope, 2010; Tang et al., 2019);  pollenkitts (which are parts of pollen surface) and SPPs (which are fragments after rupture) are slightly more hygroscopic ($0.14 \leq \kappa_{pollenkitt} \leq 0.24$, $0.1 \leq \kappa_{SPP} \leq 0.2$) (Mikhailov et al., 2019; Prisle et al., 2019; Mikhailov et al., 2020) than intact pollen grains, which can be explained by the nonuniform composition of pollen (Campos et al., 2008).

- We added the above numbers to Table 1 (please see our response to Comment 6).

***Comment 3:***

*Overall, the sensitivity studies described are useful, but there was little discussion of box model results. Specifically, more detail on the following would enhance the paper*

**Author response:** We provided more details on the model results as specified below.

*: a. lines 367-369 – why does the absorption coefficient increase at the higher wavelengths?*

**Author response (a):** We added the following explanation at line 395:

Assuming κ = 0.25 ($S_{opt10}$) instead of κ = 0.03 ($S_{opt9}$), leads to an increase of the scattering coefficient by 17 to 90% at RH = 90%. Also the absorption coefficient increases by ~40% at λ > 2 μm. This trend can be explained as the imaginary part of water is higher by three orders of magnitude at λ ~2 μm compared to that at λ ~1 μm (Kou et al., 1993). It can be concluded that the importance of $\kappa_{PBAP}$ increases at higher RH, as under these conditions PBAP hygroscopic growth is most efficient.

*b. Figure 6 – large changes with refractive indices (no surprise) but hardly any discussion in the text of what changes are important*

**Author response (b):** We discussed in more detail what changes are important. The change of optical properties within different species of bacteria (red lines) or different species of fungi (blue lines) can be larger than that between bacteria and fungi. Therefore, the detailed information about

the species of PBAPs is important in order to better model the optical properties. These differences in scattering and absorption can induce significant change in radiative forcing and will be discussed in section 4.1.4.

We modified the following at the beginning of section 4.1.3 at line 436:

The complex refractive index of PBAPs can be explained by their building blocks of various functional groups (Hill et al., 2015). Here the complex refractive indices of PBAPs are based on the measurements of *Erwinia herbicola* by Arakawa et al. (2003) and twelve other PBAPs by Hu et al. (2019); the complex refractive indices of 'other particles' in the model are the averaged values based on the volume fractions of ammonium sulfate, soot, and water (**Table 2**). The calculated scattering and absorption coefficients of the total particle population are shown in **Figure 6**. Scattering coefficients for different PBAPs vary by a factor of up to four and the absorption coefficients by a factor of up to six.

The difference of optical properties between bacteria species or fungi species can be larger than that between these two types of PBAPs. Therefore, detailed information on PBAP species is important in order to estimate their direct interaction with radiation (**Section 4.1.4**).

*c. Figure 7: why are the nitrated changes in scattering large at smaller wavelengths?*

**Response (c):** The scattering and absorption coefficients are affected by the real part and the imaginary part in non-linear ways. We modified discussion at line 453:

Due to the lack of data on the change of complex refractive index (Δm) for nitrated proteins in PBAPs, we assume nitrated PBAPs have a similar change in the refractive index to that of SOA ($S_{opt12}$ and $S_{opt13}$).  After nitration, the scattering coefficient decreases by ~20% in the range of 300 nm < λ < 450 nm and is nearly constant in the range of 460 nm < λ < 560 nm (*Figure 7a*). The scattering coefficient depends non-linearly on the real and the imaginary parts. The absorption coefficient of nitrated PBAPs is higher by 14% to 160% in the range of 300 nm < λ < 540 nm (*Figure 7b*) and is nearly constant in the range of 550 nm < λ < 560 nm. The largest difference (~160%) for absorption coefficient is observed at 440 nm and the smallest difference (~6%) is observed at 560 nm, which can be attributed to the wavelength-dependent change of the imaginary part (Δk) (Liu et al., 2015).

***Comment 4:***

*The ranking in Figure 11 is potentially useful but ultimately confusing. Please revise the accompanying text to make this figure more clear – right now the discussion is scattered and it would help to clarify this figure more, as it is ultimately very useful.*

**Responses and revisions 4:**

We thank the referee for acknowledging the value of the last figure. We modified and extended the discussion of Figure 11 in Section 5. Since Referee #2 had also major concerns about this figure, we frame its discussion now more in the context of our process model results and the need of future studies to characterize PBAP properties, rather than making strong claims about global implications. We also emphasize throughout the revised manuscript that the role of PBAP in the atmosphere for the aerosol direct and indirect effect may be limited due to their small number concentration on a global scale. However, detailed knowledge on PBAP properties that affect their interaction with radiation and water vapor is also essential to properly describe their transport, dispersion and lifetime in the atmosphere, which might affect the global modification of biodiversity and impacts public health.

As the previous title did not imply this, we changed it accordingly:

[revised manuscript text omitted]

*Minor comments: 1. The acronym used in the paper is inconsistent with the literature on \*primary\* biological aerosol particles (PBAP) not BAP. While they do talk about some secondary processing of the aerosols, the origin of the particles is still primary (as opposed to secondary formation), and consistency with prior work is helpful.*

**Responses 1:** We agree with the referee that terminology consistent with the literature should be preferred to avoid confusion. We have changed BAP to PBAP throughout the manuscript, including figures.

*2. Line 57 – is the Londahl et al. 2014 the correct reference here? This seems to be an error.*

**Responses 2:** We apologize for the confusion. It was indeed a wrong reference. We replaced it by the correct reference by the same author

*3. Line 58 – Myhre et al. 2013 is not in the reference list.*

**Responses 3:** The reference was added to the reference list.

*4. Line 98 – Pollen can also nucleate ice – see Diehl et al. 2001*

**Responses 4:** We already included pollen in *Table 1*. We discussed the ice nucleation property of pollen and modified the following sentences:

In addition to acting as CCN, some species of bacteria, fungi, and pollen can nucleate ice at high temperatures (Hoose and Möhler, 2012; Morris et al., 2004, 2008; Pouzet et al., 2017; Diehl et al., 2001, 2002), which makes them unique in terms of ice nucleation to affect the evolution of mixed-phase clouds at these temperatures (*Figure 1c*).

*5. Line 181 – missing word between "that" and "might"? Can't tell what this sentence is supposed to say*

**Responses:** Thanks for pointing out this omission. We completed the sentence as follows:

Due to the similarity of the molecular structure of organic macromolecules (e.g. proteins) and secondary organic aerosols (SOA), it can be likely assumed that nitration might alter the BAP refractive index similar to that of SOA.

*6. Table 1: missing many references on the rupture of pollen. I actually think that these numbers are incorrect and very much mis-represent the range of potential sizes (see refs Grote, Taylor, Suphioulglu for a few; listed below). Also, you are missing the rupture of fungal spores (China, Lawler; see references below). Also missing the fact that the hygroscopicity of pollen may change on rupture (not just from oxidation).*

**Responses 6:** We extended Table 1:

**Table 1.** Physicochemical properties of various PBAPs and their changes due to physical, chemical and biological ageing processes based on literature data.

[revised manuscript text omitted]

In addition, in the conclusion section at line 720, we modified the following sentences:

$\kappa_{PBAP}$ might be modified by physical (e.g., release of inner molecules due to rupture of pollen and fungal spores, condensation of gases), biological (e.g., formation of biosurfactants or other metabolic products), and chemical (e.g., nitration, oxidation) processes.

*7. Lines 283-284: The text should more clearly state that certain classes of PBAP are excluded based on the 0.5-2.8 micron size representation.*

**Responses 7:** We add at line 312:

Thus, the simulations focus on PBAPs in this size range and exclude smaller (e.g. viruses, SFPs or SPPs) and larger (e.g. pollen grains) particles.

*8. Line 342: Fungal spores could also be on the order of this size. . .*

**Responses 8:** We added at line 365:

Larger PBAPs ($D_{PBAP} = 3$ µm, $S_{opt6}$) such as SPPs and fungal spores lead to an increase in the scattering coefficient by a factor of 1.4-4.7 depending on $\lambda$.

*9. Lines 360-363: This line downplays the potential importance of non-spherical particles. The true atmospheric range of moisture conditions is not enough to say what is more likely, therefore this speculation should be removed and it would be better to discuss what types of uncertainties non-spherical particles would include.*

**Responses 9:** We removed the speculation at lines 360-363. We found some papers about the uncertainties of non-spherical particles. We added the following to the end of section 4.1.1 at line 384:

Non-sphericity of particles might translate into the same changes as caused by different particles sizes, which might induce uncertainties including optical depth and surface albedo (Kahnert et al., 2007). These uncertainties on scattering and absorption caused by non-spherical shape might be of comparable magnitude to that caused by the complex refractive index (Yi et al., 2011).

*10. Figure 3: the caption states that there is an a/b panel to capture scattering and absorption, yet only the scattering is shown.*

**Responses 10:** We apologize for the omission of Figure 3b in the original manuscript. Actually, we added Figure 3b in the supporting information (as Figure S2) We changed the Figure caption accordingly. Its information is rather limited since the absorption for all PBAPs is (nearly) identical, i.e., the absorption coefficient is not affected in the presence of PBAPs.

*11. Line 392: "very small PBAP" could also be pollen or fungal fragments. Please see literature suggestions in the major comments.*

**Responses 11:** We have changed the sentences at line 427:

Only for very small PBAPs, i.e. representative for viruses, SPPs or SFPs (***Section 2.1***), the curvature term significantly influences *s* (***Figure 5***).

*12. Line 422: I think this is supposed to be S15 and S16?*

**Responses 12:** The referee is correct; we meant to refer to S15 and S16, rather than S13 and S14. As we deleted some simulations in Table 3 to make it shorter, their number changed to S12 and S13.

*13. Line 471: what is delta_mBAP? First use, please define. (perhaps including S13, S15 and S16?)*

**Responses 13:** Delta-mBAP means the change of refractive index due to different types of BAP or nitration. We will define Delta-mBAP at line 454. S13 means simulation 13. We will define them at first use at line 316, line 331, and line 343.

*14. Table 3: last row – is dm_aged the same as dm_nitrated? Different terminology than Table 2.*

**Responses 14:** They are the same. We use now dm_nitrated for the whole manuscript.

*15. Also in Table 3 – what is the dm actually referring to? Hard to tell from comparing with Table 2.*

**Responses:** dm means the change of refractive index. We defined this at line 454. We also included more information in Table 3 (see response to Comment 3b).

*16. Lines 495-298: Could be compared with the observed values of Sc from Steiner et al. 2015*

**Responses:** We thank the referee for this additional reference. In addition, we also added data from a recent publication by Mikhailov et al. (2020) who investigated the hygroscopic behavior of various SPPs. We added the following text at line 549:

Steiner et al. (2015) reported critical supersaturations ($S_c$) of 0.81 (± 0.07)% for 50-nm SPPs and 0.26 (± 0.03)% for 100 nm SPPs. These values are similar to the values discussed above (0.68% to 1.79% for 50-nm particles, 0.24% to 0.69% for 100-nm particles) and are also in agreement with values based on the hygroscopicity ($0.1 \leq \kappa_{SPP} \leq 0.2$) reported by Mikhailov et al. (2019, 2020).

*17. Line 500 – The missed rupture literature could also be important here. Physical processes like rupture could create many more hygroscopic particles.*

**Responses:** Many thanks. We have added rupture and modified the sentences at line 547:

Thus, only for fairly small PBAPs such as viruses, SPPs and SFPs ($D \leq 100$ nm), the hygroscopicity $\kappa_{PBAP}$ may impact their CCN activation.

*18. Lines 568-571 – overall, these changes are really hard to see in the figure. Is it possible to overlay Figures 10a/b so they can be more directly compared?*

**Responses:** We intentionally separate figure 10a, 10b and 10c to show that the start of Bergeron-Findeisen process occurs at different temperatures. Namely, the curves in figure 10a, 10b, and 10c are the same whereas the y-axis shows different scales. We make this clearer now in the figure caption at line 643:

**Figure 10.** Percentage contribution of ice water content (%IWC, dashed lines) and liquid water content (%LWC, solid lines) total adiabatic water content for $\theta_{PBAP}$ of (a) 4°; (b) 20°; (c) 40° and (d) 37° and 38°. The curves in the first three panels exhibit similar shapes for different temperature ranges, i.e. the Bergeron-Findeisen process starts at different temperatures. The last panel shows that even when the contact angle increases by 1°, the temperature, at which the LWC fraction starts decreasing, differs significantly.

*19. Lines 630-632: What is the reference for this sentence "However, as it has been shown that at many locations NBAP/Ntotal is approximately constant. . .." – this is not true for fungal spores*

*and pollen. The emissions of these types of PBAP are very spatially and temporally heterogeneous, and tend to be more event-based than consistent.*

**Responses 19:** We reworded this paragraph as follows at line 690:

[revised manuscript text omitted]

*Reviewer 2:*

*This work explores the radiative effects of biological aerosol particles (BAPs). The authors conduct a literature review on the physicochemical properties associated with scattering and absorption of radiation, cloud condensation nuclei (CCN) activation and ice nucleation efficiency of BAPs. From this they establish plausible ranges for different BAPs properties, then perform several sensitivity studies to roughly assess the possible impacts on radiation and cloud evolution, hence on climate. This is a well written paper that lies within the scope of ACP. However, it is also very speculative. It is not clear that enough data has been reported to establish a possible impact. The sensitivity studies are also conducted in a very idealized and simplified way, particularly those related to cloud formation. Thus, some clarification of the approach taken, as well as the considerable limitations of this study, are required before it can be accepted for publication.*

*General Comments: The authors carry out a somehow extensive literature review to select a plausible range of parameters to carry out sensitivity studies. This is commendable; however, the process studies sprung from it may be too limited and too idealized to be meaningful regarding the effect of BAPs on radiative forcing. For example, a parcel model is not at all appropriate to make conclusions on the prevalence of the Bergeron-Findeinsen (BF) process. This is further discussed below. The process studies thus lead to somehow obvious conclusions, which are not necessarily unique to BAPs and that are already well-known, i.e., higher kappa value leads to easier CCN activation, lower contact angle to more efficient ice nucleation, in a parcel model ice grows at the expense of liquid, higher refractive index leads to enhance absorption, and so on. Thus the authors must modify the language with a honest and thorough assessment of the limitations of their study and also emphasize differences/similarities with the typical behavior of other aerosols.*

**Response:** We thank the referee for their constructive comments on our manuscript. We agree that some of the conclusions may have been too strong given the limitation of our process model studies. We have substantially revised our manuscript; the main changes include

- We changed the title to

Sensitivities to biological aerosol particle properties and ageing processes: Potential implications for aerosol-cloud interactions and optical properties

- We showed more clearly the commonalities between PBAP and other aerosol types

- we discuss in more detail the uniqueness of PBAP in terms of their modification by biological processes.

- Throughout the manuscript, and in particular in the last section, we revised the discussion of the importance of PBAP properties and their modification. We make it clearer now that their importance for radiative forcing may be limited under many conditions; however, the

properties discussed throughout the manuscript (ice nucleation and CCN activity, optical properties) should not only be explored to constrain the climatic effects but also to constrain their transport, survival and dispersion in the atmosphere. While our model framework is clearly not suited to give comprehensive estimates of all these implications, we consider our study, including Figure 11, as a useful guidance to identify the most sensitive PBAP properties and processes.

We give more details on our revisions in our point-by-point responses below.

In the abstract, because our small-scale model cannot quantify the effect of PBAP on CCN, direct radiation, and IN, we have rephrased the abstract as follows:

Primary biological aerosol particles (PBAPs) such as bacteria, viruses, fungi, and pollen, represent a small fraction of the total aerosol burden. Based on process model studies, we identify trends in the relative importance of PBAP properties, e.g. number concentration, diameter, hygroscopicity, surface tension, contact angle, for their aerosol-cloud interactions and optical properties. While the number concentration of PBAPs likely does not affect total CCN concentrations globally, small changes in the hygroscopicity of submicron PBAPs might affect their CCN ability and thus their inclusion into clouds. Given that PBAPs are highly efficient atmospheric ice nuclei at T > -10 °C, we suggest that small changes in their sizes or surface properties due to chemical, physical or biological processing might translate into large impacts on ice initiation in clouds. Predicted differences in the direct interaction of PBAPs with radiation can be equally large between different species of the same PBAP type and among different PBAP types. Our study shows that not only variability of PBAP types, but also their physical, chemical, and biological ageing processes might alter their CCN and IN activities and optical properties to affect their aerosol-cloud interactions and optical properties. While these properties and processes likely affect radiative forcing only on small spatial and temporal scales, we highlight their potential importance for PBAP survival, dispersion and transport in the atmosphere.

In addition, we largely rewrote the conclusion to stress the features that are characteristic for PBAPs (also our response to Comment 4 by Referee #1).

[revised manuscript text omitted]

*Detailed comments: Line 11. Are biological fragments considered here?*

**Response:** Also in response to Referee #1, we added more details and discussion on rupture of pollen and fungi. Accordingly, we added text in the introduction, Section 2.1 (including Table 1), and 2.3.1.

- in the introduction at line 78:

In particular pollen rupture leads to a huge increase in the number of subpollen particles (SPPs) (Bacsi et al., 2006; Suphioglu et al., 1992; Taylor et al., 2004; Wozniak et al., 2018). By assuming that one pollen grain releases up to $10^6$ SPPs, regional model studies suggested that the resulting SPPs can significantly suppress seasonal precipitation (Wozniak et al., 2018).

- At the end of Physical transformations at line 105:

For example, the break-up of pollen or fungi due to rupture can lead to higher number concentrations by several orders of magnitude (Suphioglu et al., 1992; Wozniak et al., 2018).

- We modified the text at the end of Section 2.1 at line 174:

At high RH and during precipitation or thunderstorms, pollen absorb water and one pollen grain can release $\sim 10^3$ SPPs due to osmotic pressure (Grote et al., 2001; Suphioglu et al., 1992).  This process can result in fragments with diameters of 1-4 μm and number concentrations of $N_{SPP}$ ~0.1 cm$^{-3}$ during thunderstorms (Zhang et al., 2019). These concentrations correspond to ~1 to 25 ng m$^{-3}$ ($D_{SPP} < 2$ μm) (Miguel et al., 2006). Laboratory chamber measurements have shown that SPPs from rupture of fresh birch pollen or grass pollen have diameters of in the range of 0.03 to 4.7 μm (Taylor et al., 2002, 2004). Recent laboratory measurements suggest that also fungal spores can rupture, resulting in subfungi particles (SFPs) with $D_{SFP}$ of 0.03 to 0.9 μm after exposure to high relative humidity (China et al., 2016). Ambient measurements suggest $N_{SFP}$ of 150 to 455 cm$^{-3}$ (10 nm < $D_{SFP}$ < 100 nm) after rainfall; observed peaks in aerosol size distributions at 20 nm < $D_{SFP}$ < 50 nm which frequently appeared 1.5 days after rain events were ascribed to such rupture events (Lawler et al., 2020).

- We modified the following sentences at the end of Section 2.3.1 at line 225:

The hygroscopy of pollen is similar to that of bacteria: The κ value of intact pollen grains falls into the range of $0.03 \leq \kappa_{pollen} \leq 0.17$ (Chen et al., 2019; Pope, 2010; Tang et al., 2019);

 pollenkitts (which are parts of pollen surface) and SPPs (which are fragments after rupture) are slightly more hygroscopic (0.14 ≤$\kappa_{pollenkitt}$ ≤0.24, 0.1 ≤ $\kappa_{SPP}$ ≤ 0.2) (Mikhailov et al., 2019; Prisle et al., 2019; Mikhailov et al., 2020) than intact pollen grains, which can be explained by the nonuniform composition of pollen (Campos et al., 2008).

- We also added above numbers to Table 1.

*Line 37. Delete "the"*

**Response:** We deleted 'the' before location.

*Line 41. Maybe "in the urban area of Mainz" is more appropriate.*

**Response:** We changed the text as follows at line 38, agreement with the original literature:

In the semirural area of Mainz in central Europe, the number fraction was 1-50% for particles with diameter (D) > 0.4 μm (Jaenicke, 2005).

*Line 68. In Figure 1 would absorption of solar radiation lead to a semi-direct effect?*

**Response:** The referee is correct that generally the absorption of radiation by absorbing organic molecules ('brown carbon') may contribute to the semi-direct effect. However, given the small amounts of light-absorbing material in PBAPs and small mass fraction of PBAP total absorbing mass, the global effect is likely small. In addition, the semi-direct effect is mostly triggered by light-absorbing material (e.g. soot particles) above clouds. Given the large sizes of most PBAP, their concentrations decrease strongly with altitude (Ziemba et al., 2016), thus their impact near cloud top may be small. We have added at line 469:

While generally, light-absorbing organics ('brown carbon') might contribute to the aerosol semi-direct effect (Brown et al., 2018; Hansen et al., 1997), i.e. the impact of aerosol heating on clouds, it seems unlikely that PBAPs have a significant contribution to it. Given the supermicron sizes of most PBAPs, their concentration decreases strongly as a function of altitude (Ziemba et al., 2016) and thus their concentration near cloud tops is likely negligible.

*Line 165. Must be "agglomerates"*

**Response:** We replaced 'agglomerate' to 'agglomerates'.

*Line 240. It is not clear what the maximum frozen fraction means here. If the temperature is lowered to -40 C the bacteria won't freeze at all? Why is -10 C the temperature of choice?*

**Response:** The term 'maximum frozen fraction' was misleading. We were referring to the fraction of PBAPs that have IN macromolecules. In the study by Joly et al. (2013), experiments were performed at T ≥ -10 °C. As PBAPs are the predominant atmospheric particles that nucleate ice above this temperature, we focussed in our model studies on this temperature range.

We changed the wording as follows at line 268:

For example, only 0.1 to 10% of *Pseudomonas syringae* cells express IN active macromolecules (Joly et al., 2013).

*Line 244. This is factually wrong. All real materials show stochastic behavior during ice nucleation. Please rephrase.*

**Response:** We changed the text as follows at line 263:

 However, it has been shown that bacteria of the same species and within the same population often exhibit different ice nucleation behavior (Bowers et al., 2009; Failor et al., 2017; Fall and Fall, 1998; Lindow et al., 1978; Morris et al., 2004). This behavior has been explained by various expression levels of IN-active macromolecules that are located at the cell surface. Under conditions such as phosphate starvation, the expression level might be higher, which is a strategy to reach nutrients after destroying the cells of plants by freezing (Fall and Fall, 1998). For example, only 0.1 to 10% of *Pseudomonas syringae* cells express IN active macromolecules (Joly et al., 2013). Bacteria from the same population without expression of such molecules did not freeze under the experimental conditions.

*Line 250. The application of the contact angle approach to ice nucleation in biological materials is fraught with problems, since all the assumptions of classical nucleation theory break, and depends strongly on the values selected for other very uncertain parameters like for example the ice-liquid interfacial tension and the activation energy. Please add an explanation on the limitations of describing ice nucleation in biological materials.*

**Response:** The referee is correct that the contact angle should be regarded as a fitting parameter, rather than as a physicochemical parameter, exactly describing the IN surface.

The implications of different expressions for the activation energy, germ formation and other factors included in the classical nucleation theory have been discussed in detail before (Hoose and

Möhler, 2012). We added the reference of Ervens and Feingold (2012) where the detailed model description is given.

We modified at the beginning of section 2.4.2 at line 272:

In agreement with previous studies, we base our discussion on the contact angle as a fitting parameter in the classical nucleation theory (CNT) to parametrize the frozen fraction observed in experiments. In agreement with previous studies, we base our discussion on the contact angle as a fitting parameter in the classical nucleation theory (CNT) to parametrize the frozen fraction observed in experiments. If not reported in the respective experimental studies, we assumed a freezing time of 10 seconds to derive $\theta$ from experimental data, in agreement with many experimental conditions (Attard et al., 2012; Gute and Abbatt, 2018; Kunert et al., 2019). All CNT model equations and parameters are identical to those as described by Ervens and Feingold (2012); Hoose and Möhler (2012) discussed different assumptions made for the various variables in the CNT in previous ice nucleation studies.

*Line 258. INAS is obtained by fitting freezing experiments neglecting the time dependency of ice nucleation. Please rephrase. I would suggest the authors refrain from discussing deterministic vs stochastic behavior since it is distracting and not at all clear what they mean, particularly for BAPs.*

**Response:** We agree with the referee that the mentioning of deterministic behaviour is rather distracting at this place. We deleted the text in 255 – 258 and reworded the sentence as follows at line 283:

 Hoose and Möhler (2012) reported the ice nucleation active surface site (INAS) density of various bacteria at -5 °C ($10^{2.5}$-$10^{10}$ m$^{-2}$). Using CNT, we fitted a contact angle to their data, resulting in the range of 32° ≤ $\theta_{bacteria}$ ≤ 34°.

*Line 273. Are these changes due to denaturation or are they reversible?*

**Response:** Attard et al. (2012) did not investigate whether or not the observed pH effect was reversible. Based on other studies (Schmid et al., 1997; Turner et al., 1990), it can be concluded that denaturation of IN protein's agglomerates (polymers) occurs at pH below 4.5, indicating that IN activities are supposed to be reversible at least above pH 4.5. We add at the end of section 2.4.2 at line 654:

Denaturation of IN protein's agglomerates (polymers) occurs at pH below 4.5 (Schmid et al., 1997; Turner et al., 1990), suggesting that changes in IN activities due to pH might reversible at least above this pH value.

*Line 286. Is there a reason to consider BAPs externally-mixed and monodisperse?*

**Response:** The reason for considering PBAPs as being externally mixed and monodisperse is the simplicity of our model studies. We do not attempt to give quantitative estimates of their radiative forcing in the climate system, but our model sensitivity studies are set up such that we compare results from different model simulations to each other, in order to conclude on the sensitivities to individual aerosol properties. Assuming different PBAP properties such as polydisperse size distributions or internally mixed aerosol might change the numbers shown in our figures but not the relative changes due to the variation of one aerosol parameter at a time. We clarify this at line 134:

By means of process models (***Section 3***), we explore in a simplistic way the relative importance of these PBAP properties and ageing processes for the effects depicted in ***Figure 1*** (***Section 4***). Our model sensitivity studies are set up such that we identify trends and their relative importance to show the sensitivities to individual properties and ageing processes that impact PBAP properties in the atmosphere.

In addition, we frame the discussion now more in the context of our process model results and the need of future studies to characterize PBAP properties, rather than making strong claims about global implications. We also emphasize throughout the manuscript that the role of PBAP in the atmosphere for the aerosol direct and indirect effect may be limited due to their small number concentration on a global scale. However, detailed knowledge on PBAP properties that affect their interaction with radiation and water vapor is also essential to properly describe their transport, dispersion and lifetime in the atmosphere, which might affect the global modification of biodiversity and impacts public health. We added this in Section 5 (Conclusions) at line 666:

Given the limitations of our process models in terms of scales, dimensions and parameter spaces, our results should be considered as qualitative, rather than quantitative estimates; the focus of our study is the comparison of relative changes due to various physicochemical parameters.

*Line 301. What are the properties of the "other" aerosol. Is there any sensitivity of the results to this assumption?*

**Response:** The detailed properties of 'other particles' are listed in Table 2. We used the typical conditions to represent 'other particles', i.e., the majority of typical atmospheric aerosol populations. Since sensitivities on the properties of CCN activation to other aerosol types have been extensively studied, e.g., (Ervens et al., 2005; McFiggans et al., 2005), we did not consider

variation of the properties of 'other aerosol' in the current study. Just as stated in our response to the previous comment, the absolute numbers in our figures may change depending on the type and properties of the 'other aerosol', however, the general conclusions on the relative changes will likely not change.

*Line 311. If the BAPs freeze by immersion, shouldn't they be inside the droplets? Are the results sensitive to Nother?*

**Response:** In the parcel model, PBAPs first act as CCN on which droplets form. PBAPs are inside the droplets, and then immersion freezing occurs at freezing temperature. The number of other particles might affect $N_{CCN}$ and supersaturation, which in turn affects ice formation. We performed a sensitivity test of the ratio of IWC/LWC to $N_{CCN}$ in our previous study (Ervens et al., 2011), where we concluded that NCCN has likely a small impact in mixed phase clouds.

*Line 313. This is a crude approximation that only works to make an assessment on droplet/ice formation, but would be very misleading to estimate LWC and IWC. Once ice is formed a whole set of other microphysical processes rapidly take place. Please justify why this approach is used at all.*

**Response:** We agree with the referee that parcel models are of limited value in describing the full evolution of mixed-phase clouds upon the initiation of the Bergeron-Findeisen process, i.e., the full glaciation process followed by precipitation and demise of the cloud. However, they have been proven as useful tools for sensitivity studies that explored the onset of the Bergeron-Findeisen process for various aspects of ice nucleation (Diehl et al., 2006; Eidhammer et al., 2009; Ervens et al., 2011; Khvorostyanov and Curry, 2005; Korolev, 2007; Korolev and Isaac, 2003).

We add the references above and briefly discuss the limitations of the adiabatiac model framework at line 621:

It should be noted that our adiabatic parcel model framework cannot fully represent the complexity of all processes occuring in mixed-phase clouds, such as complete glaciation followed by precipitation and demise of the cloud. However, we rather demonstrate the relative changes in percentage contribution of ice water content (%IWC, solid lines) and liquid water content (%LWC, dashed lines) to total adiabatic water content near the onset of ice nucleation. Thus, we apply our model in a similar way as in previous parcel model studies that explored the onset of the Bergeron-Findeisen process to various aspects of ice nucleation (Diehl et al., 2006; Eidhammer et al., 2009; Ervens et al., 2011; Khvorostyanov and Curry, 2005; Korolev, 2007; Korolev and Isaac, 2003).

In addition, we also modified texts at line 659:

Overall,  our model results suggest that a small change in the contact angle due to different types of PBAPs or due to ageing processes might have a large impact on ice nucleation in clouds  These differences might translate into feedbacks on other subgrid and dynamical processes in the cloud that amplify or reduce the efficiency of glaciation. However, such processes cannot be further explored in the adiabatic parcel model framework.

*Line 330. This "Nother" is different from the "Nother" of line 309, which is also different to the one in line 301.*

**Response:** We agree that it was confusing to use the identical name 'Nother' in three different contexts. We now distinguish the three values of Nother and indicate that they are used in the simulations of CCN, IN and optical properties, respectively:

In line 328: The dry aerosol size distribution covers a size range of 5 nm < $D_{other,\ S(CCN)}$ < 7.7 μm with $N_{other,\ S(CCN)}$ = 902 cm$^{-3}$, as being typical for moderately polluted continental conditions.

In line 336: We consider an aerosol size distribution with 46 nm < $D_{other,\ S(IN)}$ < 2.48 μm in nine size classes and $N_{other,S(IN)}$ = 100 cm$^{-3}$, as found in Arctic mixed-phase clouds. The aerosol population includes one additional PBAP size class, which is the only one that includes potentially freezing IN under the model conditions.

In line 354: Note that the concentration of other particles ($N_{other,\ S(opt)}$) would usually increase under haze conditions while we keep $N_{other,\ S(opt)}$ as a constant in the above model (1.4 cm$^{-3}$);

*Line 438. All of these values change strongly with location, so it is not clear why this estimate is not given with a range of uncertainty, down in line 450.*

**Response:** The referee is correct that all values in Eq-5 are strongly time and location dependent. However, we clarify that this estimate is only intended to compare in a relative sense the RFE due to differences in optical properties. We adapted this approach including all values in Eq-5 from Dinar et al. (2007). We also clarified that our results are obtained for relative comparisons, rather than for general or global of radiative forcing calculations at line 500:

The RFE values in ***Table 3*** only represent radiative forcing of a small range of particle sizes and a constant composition and number concentration of other particles; however, the differences (ΔRFE) allow evaluating the relative importance of the various PBAP parameters ($N_{PBAP}$, $D_{PBAP}$, $m_{PBAP}$) in terms of their direct interaction with radiation. A negative ΔRFE implies more scattering and a positive ΔRFE implies more absorption due to the presence of PBAPs.

Note that in the above simulations relatively high concentrations of PBPAs were assumed and should only be used to compare the relative importance of PBAP size and complex refractive index

for their optical properties. The properties of PBAPs can vary depending on species of PBAPs and ageing processes. Given that the number concentration of PBAPs is generally small, the direct radiative effect of PBAPs is likely restricted to small spatial scales.

*Line 459. This would only be true if BAPs were uniform in the globe and isolated from other aerosols.*

**Response:** See our response to the previous comment. We hope that our text changes and additions above are sufficient to clarify that our intention not to simulate the global effect of PBAPs. Our main idea is to see the difference of RFE ($\Delta$RFE) induced by the addition of PBAPs in a relative sense.

*Line 479. There is certainly not data to support this "independence" assertion. The authors could probably make this assumption but clarify that it is in the absence of better data.*

**Response:** We found more data, added more references, and reworded this paragraph as follows at line 690 (also response 19 to reviewer #1):

The number fraction of PBAPs to total CCN is relatively small ($\leq$ ~0.1%). For example, in the Amazon, it is on the order of 0.01 to 0.1% based on the reported ranges of PBAP number concentrations ($0.2 < N_{PBAP} < 1.2$ cm$^{-3}$ (Whitehead et al., 2016); $0.04 < N_{PBAP} < 0.13$ cm$^{-3}$ (Huffman et al., 2012)) and CCN concentration ($N_{CCN}$ ~260 cm$^{-3}$, at 1% supersaturation (Roberts et al., 2001)). A similar ratio of $N_{PBAP}/N_{CCN}$ (~0.01 to 0.1%) can be derived based on measurements in the megacity Beijing with $N_{PBAP} \leq 1.4$ cm$^{-3}$ (Wei et al., 2016) during haze days and $N_{CCN} \leq 9.9 \cdot 10^3$ cm$^{-3}$ (at 0.86% supersaturation) (Gunthe et al., 2011). Thus, a small change in $N_{PBAP}$ likely does not significantly affect cloud droplet number concentration. Only in rare events, e.g. when pollen grains rupture with high efficiency, $N_{pollen}$ might considerably affect $N_{CCN}$ (Wozniak et al., 2018). However, droplet formation on PBAPs increases microorganisms' survival rate and decreases their atmospheric residence time due to precipitation, so the knowledge of their CCN-relevant properties is of biological relevance.

*Line 509. I am not sure what is shown here. This caption needs more information, Table 2 does not even say what Senv or Sc are.*

**Response:** We clarified the caption as follows at line 585:

***Figure 8***. Comparison of the environmental supersaturation within the cloud ($S_{env}$) as predicted by the parcel model for different updraft velocities (w) to the critical supersaturation ($S_c$) of PBAPs based on Köhler theory. Results are shown as a function of (a) hygroscopicity parameters $\kappa_{PBAP}$ and (b) surface tension $\sigma_{PBAP}$. Input parameters to the parcel model are listed in ***Table 2***.

*Line 531. I don't think this is a buffer effect, or at least explain what that means in this context.*

**Response:** We removed the word buffering as it may require more definition in this context and may cause confusion as we mostly focus on physicochemical aerosol properties. We intended to use it in the same context as by Feingold and Stevens (2009) who introduced this term to describe the lower sensitivity of cloud properties to aerosol characteristics in the complex aerosol-cloud systems than it is usually suggested if individual aerosol processes or properties were considered separately. We changed the text as follows at line 580:

Our sensitivity studies show  relatively lower sensitivity of cloud properties to particle composition than that predicted based on equilibrium conditions, in agreement with previous sensitivity studies (Ervens et al., 2005). Therefore, previous estimates of surfactant effects on cloud properties that are based on a simplified assumption of equilibrium conditions in clouds (Facchini et al., 1999), led to an overestimate of the role of surfactants on CCN.

*Line 561. This conclusion is short-sighted. D influences droplet activation hence where freezing could occur. Mixed-phase clouds are CCN limited as well, so the effect may not be negligible.*

**Response:** We will rephrase the sentence and also refer to the discussion of CCN properties in order to make it clear that immersion freezing is both a function of CCN and IN properties at line 617:

 For SPPs and SFPs with D ≤ 100 nm, immersion freezing may be limited by the droplet formation on these particles (***Figure S3***). As ice formation is less efficient on non-activated particles ('condensation freezing'), the onset temperatures of freezing is significantly lower. As supermicron particles likely act as CCN under most conditions, this limitation might be smaller for large PBAPs.

In addition, we add a Figure S3 to the supplement:

[Figure]

**Figure S3.** Percentage contribution of ice water content (IWC, dashed lines) and liquid water content (LWC, solid lines) to total adiabatic water content as a function of $D_{PBAP}$.

*Line 570. Please explain how the authors pin the BF effect to a particular Delta_T (also what Delta_T means). Is the T shift related to a later onset of freezing?*

**Response:** We define the onset of the Bergeron-Findeisen process as the point at which LWC% starts to decrease. Accordingly, we can compare the temperatures at which this occurs between the different simulation and $\Delta T$ means the change of temperatures of the onset of BF processes due to the change of contact angle.

The T shift is related to the point at which LWC% starts to decrease.

We have added the following at line 606:

We define the onset of the Bergeron-Findeisen process as the temperature, at which the liquid water content fraction starts to efficiently decrease.

We have changed the texts as follows at line 636:

PBAPs exhibit a wide range of contact angles of $4° < \theta_{PBAP} < 44°$ (***Table 1***). ***Figure 10*** compares the predicted relative contributions of %IWC and %LWC to the total adiabatic water content. The comparison of Figures 10a and 10b shows that the onset temperatures of the %LWC decrease are at ~ -7.7 °C ($\theta_{PBAP} = 4°$) and ~ -8.3 °C ($\theta_{PBAP} = 20°$), respectively, i.e. resulting in a difference of $\Delta T$ ~0.6 °C. This difference is predicted to be larger ($\Delta T$ ~3.3 °C) for PBAPs with $\theta_{PBAP} = 40°$.

*Line 574. No, this is not clear at all. Early freezing may result in early scavenging of available BAP and actually limiting instead of enhancing BF processes. There is a myriad of other things that can negate the onset of BF process, none of which can be represented in a parcel model: high subgrid scale vertical velocity, the presence of other efficient ice nucleating particles (for example feldspards can freeze at very high T as well), preferential spatial concentration of liquid and ice particles, to name a few. I would accept a much more cautious language like for example, "has the potential to affect the BF process" followed by a list of all the things that need to be addressed before this conclusion can be asserted with any degree of accuracy.*

**Response:** We changed the language to more cautious and talked about limitations of adiabatic parcel models (see also our response to the previous comment). Although the feldspars can freeze at very high T, the nucleation site density of feldspars is much lower than the bacteria. We modified the discussion as follows at line 646:

As discussed in **Section 2**, chemical (e.g., nitration, oxidation, adjustments due to pH) or physical processing of IN surfaces might lead to $\Delta\theta_{PBAP} \sim 1°$. In **Figure 10d**, we show %IWC and %LWC by comparing $S_{IN2}$ and $S_{IN9}$. The results show that even such a small change of 1° in $\theta$ can cause a significant difference in the predicted IWC and LWC evolutions. The temperature, at which the %LWC starts decreasing differs by $\Delta T \sim 1.3$ °C. Such a change in $\theta$ may be induced by pH changes; for example, it was found that $\Delta\theta$ is $\sim 1.5°$ for bacteria such as *Pseudomonas syringae* when the cells were exposed to solutions of pH 7.0 and 4.1 at temperatures of T > -10 °C. Denaturation of IN protein's agglomerates (polymers) occurs at pH below 4.5 (Schmid et al., 1997; Turner et al., 1990), suggesting that changes in IN activities due to pH might be reversible at least above this pH value.

Similar differences in $\theta$ could be also caused due to other processes, such as the oxidation of pollen that lead to $\Delta\theta \sim 1.5°$ at T $\sim$ -39 °C (Gute and Abbatt, 2018). However, at this much lower temperature, the sensitivity of the frozen fraction to $\Delta\theta$ decreases (Ervens and Feingold, 2013). Overall, our model results suggest that a small change in the contact angle due to different types of PBAPs or due to ageing processes might have a large impact on ice nucleation in clouds. These differences might translate into feedbacks on other subgrid and dynamical processes in the cloud that amplify or reduce the efficiency of glaciation. However, such processes cannot be further explored in the adiabatic parcel model framework.

*Line 584. Please show here where the BF process is initiated.*

**Response:** We rephrased the texts about BF process initiation at line 643:

**Figure 10.** Percentage contribution of ice water content (%IWC, dashed lines) and liquid water content (%LWC, solid lines) total adiabatic water content for $\theta_{PBAP}$ of (a) 4°; (b) 20°; (c) 40° and

(d) 37° and 38°. The curves in the first three panels exhibit similar shapes for different temperature ranges, i.e. the Bergeron-Findeisen process starts at different temperatures. The last panel shows that even when the contact angle increases by 1°, the temperature, at which the %LWC fraction starts decreasing, differs significantly.

*Line 596. Figure 11 must be removed. To start it is confusing since clearly the different aging processes affect more than one variable at a time. More fundamentally it presents a misleading, "final" assessment of something that is highly uncertain. The data is still too scarce and the studies way too idealized to support this figure.*

**Response:** We agree with the referee that our conclusions based on our model results and describing this figure may have been too strong as our limited process model studies should not be extended to the global scale. However, we would like to keep this figure as it is to our knowledge the first overview of the potential role of biological processes that may affect PBAP properties in the atmosphere. Instead of framing it in the context of radiative forcing, we now focus more on the measurement needs of PBAP properties and processes and their potential (limited) influence on radiative forcing.

We frame its discussion now more in the context of our process model results and the need of future studies to characterize PBAP properties, rather than making strong claims about global implications. We also emphasize throughout the manuscript that the role of PBAPs in the atmosphere for the aerosol direct and indirect effect may be limited due to their small number concentration on a global scale. However, detailed knowledge on PBAP properties that affect their interaction with radiation and water vapor is also essential to properly describe their transport, dispersion and lifetime in the atmosphere, which might affect the global modification of biodiversity and impacts public health.

We rewrote Section 5 (see reply to general comments).

We also modified Figure 11 and caption as follows:

[Figure]

**Figure 11**. Schematic of PBAP types and ageing processes that affect their aerosol-cloud interactions and optical properties. The bottom arrow shows the increasing fraction of $N_{PBAP}$ to total particles ($N_{CCN}$, $N_{> 5\ \mu m}$, and $N_{IN}$, respectively)**.** The left arrow indicates the increasing sensitivity to PBAP properties as predicted based on our process model studies. The various properties might be modified by physical (green), chemical (blue) and biological (red) ageing processes.

*Line 615. What about the semi-direct effect?*

**Response:** The referee is correct that generally the absorption of radiation by absorbing organic molecules ('brown carbon') may contribute to the semi-direct effect. However, given the small amounts of light-absorbing material in PBAPs and small mass fraction of PBAP total absorbing mass, the global effect is likely small. In addition, the semi-direct effect is mostly triggered by light-absorbing material (e.g. soot particles) above clouds. Given the large sizes of most PBAP, their concentrations decreases strongly with altitude (Ziemba et al., 2016), thus their impact near cloud top may be small. We have added at line 469:

While generally, light-absorbing organics ('brown carbon') might contribute to the aerosol semi-direct effect (Brown et al., 2018; Hansen et al., 1997), i.e. the impact of aerosol heating on clouds, it seems unlikely that PBAPs significantly have this effect. Given the submicron sizes of most PBAPs, their concentration decreases strongly as a function of altitude (Ziemba et al., 2016) and thus their concentration near cloud tops is likely negligible.

*Line 630. See comment on Line 479.*

**Response:** We found more data, added more references, and reworded this paragraph as follows at line 690 (also response 19 to reviewer #1):

The number fraction of PBAPs to total CCN is relatively small ($\leq$ ~0.1%). For example, in the Amazon, it is on the order of 0.01 to 0.1% based on the reported ranges of PBAP number concentrations ($0.2 < N_{PBAP} < 1.2$ cm$^{-3}$ (Whitehead et al., 2016); $0.04 < N_{PBAP} < 0.13$ cm$^{-3}$ (Huffman et al., 2012)) and CCN concentration ($N_{CCN}$ ~260 cm$^{-3}$, at 1% supersaturation (Roberts et al., 2001)). A similar ratio of $N_{PBAP}/N_{CCN}$ (~0.01 to 0.1%) can be derived based on measurements in the megacity Beijing with $N_{PBAP} \leq 1.4$ cm$^{-3}$ (Wei et al., 2016) during haze days and $N_{CCN} \leq 9.9 \cdot 10^3$ cm$^{-3}$ (at 0.86% supersaturation) (Gunthe et al., 2011). Thus, a small change in $N_{PBAP}$ likely does not significantly affect cloud droplet number concentration. Only in rare events, e.g. when pollen grains rupture with high efficiency, $N_{pollen}$ might considerably affect $N_{CCN}$ (Wozniak et al., 2018). However, droplet formation on PBAPs increases microorganisms' survival rate and decreases their atmospheric residence time due to precipitation, so the knowledge of their CCN-relevant properties is of biological relevance.

*Line 636. See comment on Line 240.*

**Response:** The maximum frozen fraction is misleading. We were referring to the fraction of PBAPs that have IN macromolecules. We modified the sentence at 709:

Lab measurements have shown that up to 100% of pollen grains have IN nucleating macromolecules on their surface, whereas only 0.01 to 10% of bacteria express the proteins or other macromolecules that initiate ice nucleation (Failor et al., 2017; Joly et al., 2013; Pummer et al., 2015).

*Line 654. This is speculation, since the authors do not perform any studies on cell generation.*

**Response:** We use the term 'cell generation' here in the same way as in our previous study where we referred to it as the combination of cell growth and multiplication (Ervens and Amato, 2020), in agreement with the literature on bacterial processes (Marr, 1991; Price and Sowers, 2004; Si et al., 2017). In this previous exploratory study, we performed an estimate of the potential role of cell generation (i.e. focusing on increase of cell size as we were only concerned with the increase in biological mass) during the atmospheric residence time of a bacteria cell. While the growth of an individual bacteria cell cannot be monitored during its time in the atmosphere, there are several studies that support the hypothesis of growth, metabolic activity and possibly multiplication of cells in the atmosphere (Marr, 1991; Middelboe, 2000; Price and Sowers, 2004; Sattler et al., 2001; Vrede et al., 2002).

The referee is correct that to date, any conclusion on the extent to which such processes affect PBAP properties are speculative. Indeed, they are not comprehensively or at all explored yet in atmospheric models due to the lack of suitable data sets. This lack of knowledge is one of our main reasons to keep Figure 11 in the manuscript as we hope that it may initiate field, lab and model studies. Data from such studies will help to identify the most important processes that modify PBAP radiative properties and adaptive strategies of microorganisms in the atmosphere.

References 2:

Attard, E., Yang, H., Delort, A. M., Amato, P., Pöschl, U., Glaux, C., Koop, T. and Morris, C. E.: Effects of atmospheric conditions on ice nucleation activity of Pseudomonas, Atmos. Chem. Phys., 12(22), 10667–10677, doi:10.5194/acp-12-10667-2012, 2012.

Bacsi, A., Choudhury, B. K., Dharajiya, N., Sur, S. and Boldogh, I.: Subpollen particles: Carriers of allergenic proteins and oxidases, J. Allergy Clin. Immunol., doi:10.1016/j.jaci.2006.07.006, 2006.

Bowers, R. M., Lauber, C. L., Wiedinmyer, C., Hamady, M., Hallar, A. G., Fall, R., Knight, R. and Fierer, N.: Characterization of airborne microbial communities at a high-elevation site and their potential to act as atmospheric ice nuclei, Appl. Environ. Microbiol., doi:10.1128/AEM.00447-09, 2009.

Brown, H., Liu, X., Feng, Y., Jiang, Y., Wu, M., Lu, Z., Wu, C., Murphy, S. and Pokhrel, R.: Radiative effect and climate impacts of brown carbon with the Community Atmosphere Model (CAM5), Atmos. Chem. Phys., doi:10.5194/acp-18-17745-2018, 2018.

Campos, M. G. R., Bogdanov, S., de Almeida-Muradian, L. B., Szczesna, T., Mancebo, Y., Frigerio, C. and Ferreira, F.: Pollen composition and standardisation of analytical methods, J. Apic. Res., doi:10.1080/00218839.2008.11101443, 2008.

China, S., Wang, B., Weis, J., Rizzo, L., Brito, J., Cirino, G. G., Kovarik, L., Artaxo, P., Gilles, M. K. and Laskin, A.: Rupturing of biological spores as a source of secondary particles in Amazonia, Environ. Sci. Technol., doi:10.1021/acs.est.6b02896, 2016.

Coluzza, I., Creamean, J., Rossi, M. J., Wex, H., Alpert, P. A., Bianco, V., Boose, Y., Dellago, C., Felgitsch, L., Fröhlich-Nowoisky, J., Herrmann, H., Jungblut, S., Kanji, Z. A., Menzl, G., Moffett, B., Moritz, C., Mutzel, A., Pöschl, U., Schauperl, M., Scheel, J., Stopelli, E., Stratmann, F., Grothe, H. and Schmale, D. G.: Perspectives on the future of ice nucleation research: Research needs and Unanswered questions identified from two international workshops, Atmosphere (Basel)., 8(8), doi:10.3390/atmos8080138, 2017.

Després, V., Huffman, J. A., Burrows, S. M., Hoose, C., Safatov, A., Buryak, G., Fröhlich-Nowoisky, J., Elbert, W., Andreae, M., Pöschl, U. and Jaenicke, R.: Primary biological aerosol particles in the atmosphere: a review, Tellus B Chem. Phys. Meteorol., 64(1), 15598, doi:10.3402/tellusb.v64i0.15598, 2012.

Diehl, K., Simmel, M. and Wurzler, S.: Numerical sensitivity studies on the impact of aerosol properties and drop freezing modes on the glaciation, microphysics, and dynamics of clouds, J.

Geophys. Res., 111(D7), D07202, doi:10.1029/2005jd005884, 2006.

[revised manuscript text omitted]

Taylor, P. E., Flagan, R. C., Miguel, A. G., Valenta, R. and Glovsky, M. M.: Birch pollen rupture and the release of aerosols of respirable allergens, Clin. Exp. Allergy, doi:10.1111/j.1365-

2222.2004.02078.x, 2004.

Tong, Y. and Lighthart, B.: The annual bacterial particle concentration and size distribution in the ambient atmosphere in a rural area of the Willamette Valley, Oregon, Aerosol Sci. Technol., doi:10.1080/027868200303533, 2000.

Turner, M. A., Arellano, F. and Kozloff, L. M.: Three separate classes of bacterial ice nucleation structures, J. Bacteriol., doi:10.1128/jb.172.5.2521-2526.1990, 1990.

Vrede, K., Heldal, M., Norland, S. and Bratbak, G.: Elemental composition (C, N, P) and cell volume of exponentially growing and nutrient-limited bacterioplankton, Appl. Environ. Microbiol., doi:10.1128/AEM.68.6.2965-2971.2002, 2002.

Whitehead, J. D., Darbyshire, E., Brito, J., Barbosa, H. M. J., Crawford, I., Stern, R., Gallagher, M. W., Kaye, P. H., Allan, J. D., Coe, H., Artaxo, P. and McFiggans, G.: Biogenic cloud nuclei in the central Amazon during the transition from wet to dry season, Atmos. Chem. Phys., doi:10.5194/acp-16-9727-2016, 2016.

Wirgot, N., Vinatier, V., Deguillaume, L., Sancelme, M. and Delort, A.-. M.: H2O2 modulates the energetic metabolism of the cloud microbiome, Atmos Chem Phys, 17, doi:10.5194/acp-17-14841-2017, 2017.

Wozniak, M. C., Steiner, A. L. and Solmon, F.: Pollen Rupture and Its Impact on Precipitation in Clean Continental Conditions, Geophys. Res. Lett., 45(14), 7156–7164, doi:10.1029/2018GL077692, 2018.

Zhang, M., Klimach, T., Ma, N., Könemann, T., Pöhlker, C., Wang, Z., Kuhn, U., Scheck, N., Pöschl, U., Su, H. and Cheng, Y.: Size-Resolved Single-Particle Fluorescence Spectrometer for Real-Time Analysis of Bioaerosols: Laboratory Evaluation and Atmospheric Measurements, Environ. Sci. Technol., 53(22), 13257–13264, doi:10.1021/acs.est.9b01862, 2019.

Ziemba, L. D., Beyersdorf, A. J., Chen, G., Corr, C. A., Crumeyrolle, S. N., Diskin, G., Hudgins, C., Martin, R., Mikoviny, T., Moore, R., Shook, M., Lee Thornhill, K., Winstead, E. L., Wisthaler, A. and Anderson, B. E.: Airborne observations of bioaerosol over the Southeast United States using a Wideband Integrated Bioaerosol Sensor, J. Geophys. Res., doi:10.1002/2015JD024669, 2016.

---

## Author Response (AR1)

General comments: Overall, this paper is a useful study that investigates the relevant optical properties of biological aerosol particles. They provide some excellent comparison tests of which parameters and processes are important, and provide a framework for understanding these findings.

**Major comments:**

**Referee Comment 1:**

The authors frequently talk about how they do not intend this paper to be a comprehensive literature review (e.g., lines 137-140), yet it is still important that they cover the range of values that are found in the literature. Specifically, I would like to see an inclusion of more up to date information on pollen and fungal spore rupture (see next comment)

**Responses and revisions 1:**

We thank the referee for this suggestion and agree that some discussion on pollen and fungal spore rupture should be included as an additional process. We added information on the ranges of sizes of fragments of pollen and fungal spores to the revised manuscript and indicated these changes in the following comments below.

**Comment 2:**

The authors do not provide equal weight to the physical ageing via rupture of biological particles such as fungal spores and pollen. Physical ageing processes are noted, but they have not done the appropriate literature review to accurately capture how some types of biological particles may change. This represents an important atmospheric secondary process that can change both the size distribution as well as potentially the optical properties. This should be mentioned in the introduction when discussing "physical transformations" around line 100, and more specifically throughout the paper, particularly for including observed size distributions and their influence the optical properties. Pollen rupture is mentioned briefly on lines 168-169 and as a single referenced line item in Table 1, but this underestimates this process based on the long list of epidemiological literature on this process (e.g., Suphiolglu et al. 1992; Grote et al., 2001; Taylor et al. 2002; Taylor et al. 2004). More recently, fungal spores have been shown to rupture as well (Lawler et al., 2020; China et al., 2017), and this has not been mentioned at all in the text nor in Table 1. Overall, the authors spend a lot of time on the chemical processing (e.g., nitration) and its impacts, but very little on this physical process.

**Responses and revisions 2:**

We thank the referee for pointing us to these references on the rupture of pollen and fungal spores as it contributes to ageing processes of pollen and fugal spores.

We add the following text:

- in the introduction at line 78:

In particular pollen rupture leads to a huge increase in the number of subpollen particles (SPPs) (Bacsi et al., 2006; Suphioglu et al., 1992; Taylor et al., 2004; Wozniak et al., 2018). By assuming that one pollen grain releases up to 106 SPPs, regional model studies suggested that the resulting SPPs can significantly suppress seasonal precipitation (Wozniak et al., 2018).

- at the end of 'Physical transformations' at line 105:

For example, the break-up of pollen or fungi due to rupture can lead to higher number concentrations by several orders of magnitude (Suphioglu et al., 1992; Wozniak et al., 2018).

- At the end of Section 2.1, we modified the text as follows at line 174:

At high RH and during precipitation or thunderstorms, pollen absorb water and one pollen grain can release ~ $10^3$  SPPs due to osmotic pressure (Grote et al., 2001; Suphioglu et al., 1992). Similarly, a biologically-driven physical processes might lead to enhancement of NBAP as it has been observed that pollen ruptures into This process can result in fragments with diameters of 1-4 µm and number concentrations of NSPP ~0.1 cm-3 during thunderstorms (Zhang et al., 2019). These concentrations correspond to ~1 to 25 ng m-3 (DSPP < 2 µm) (Miguel et al., 2006). Laboratory chamber measurements have shown that SPPs from rupture of fresh birch pollen or grass pollen have diameters of in the range of 0.03 to 4.7 µm (Taylor et al., 2002, 2004). Recent laboratory measurements suggest that also fungal spores can rupture, resulting in subfungi particles (SFPs) with DSFP of 0.03 to 0.9 µm after exposure to high relative humidity (China et al., 2016). Ambient measurements suggest NSFP of 150 to 455 cm-3 (10 nm < DSFP < 100 nm) after rainfall; observed peaks in aerosol size distributions at 20 nm < DSFP < 50 nm which frequently appeared 1.5 days after rain events were ascribed to such rupture events (Lawler et al., 2020).

- We modified the following sentences at the end of Section 2.3.1 at line 225:

The hygroscopity of pollen is similar to that of bacteria: The  $\kappa$  value of intact pollen grains falls into the range of  $0.03 \leq \kappa_{pollen} \leq 0.17$  (Chen et al., 2019; Pope, 2010; Tang et al., 2019); in agreement with  $\kappa$  of pollen kitts on the surface of pollen pollenkitts (which are parts of pollen surface) and SPPs (which are fragments after rupture) are slightly more hygroscopic (0.14  $\leq \kappa_{pollenkitt} \leq 0.24, 0.1 \leq \kappa_{SPP} \leq 0.2$ ) (Mikhailov et al., 2019; Prisle et al., 2019; Mikhailov et al., 2020) than intact pollen grains, which can be explained by the nonuniform composition of pollen (Campos et al., 2008).

- We added the above numbers to Table 1 (please see our response to Comment 6).

**Comment 3:**

Overall, the sensitivity studies described are useful, but there was little discussion of box model results. Specifically, more detail on the following would enhance the paper

Author response: We provided more details on the model results as specified below.

: a. lines 367-369 – why does the absorption coefficient increase at the higher wavelengths?

Author response (a): We added the following explanation at line 395:

Assuming  $\kappa = 0.25$  (Sopt10) instead of  $\kappa = 0.03$  (Sopt9), leads to an increase of the scattering coefficient by 17 to 90% at RH = 90%. Also the absorption coefficient increases by ~40% at  $\lambda > 2$  µm. This trend can be explained as the imaginary part of water is higher by three orders of magnitude at  $\lambda \sim 2$  µm compared to that at  $\lambda \sim 1$  µm (Kou et al., 1993). It can be concluded that the importance of  $\kappa_{PBAP}$  increases at higher RH, as under these conditions PBAP hygroscopic growth is most efficient.

*b.* Figure 6 – large changes with refractive indices (no surprise) but hardly any discussion in the text of what changes are important

**Author response (b):** We discussed in more detail what changes are important. The change of optical properties within different species of bacteria (red lines) or different species of fungi (blue lines) can be larger than that between bacteria and fungi. Therefore, the detailed information about

the species of PBAPs is important in order to better model the optical properties. These differences in scattering and absorption can induce significant change in radiative forcing and will be discussed in section 4.1.4.

**We modified the following at the beginning of section 4.1.3 at line 436:**

The complex refractive index of PBAPs can be explained by their building blocks of various functional groups (Hill et al., 2015). Here the complex refractive indices of PBAPs are based on the measurements of *Erwinia herbicola* by Arakawa et al. (2003) and twelve other PBAPs by Hu et al. (2019); the complex refractive indices of 'other particles' in the model are the averaged values based on the volume fractions of ammonium sulfate, soot, and water (*Table 2*). The calculated scattering and absorption coefficients of the total particle population are shown in *Figure 6*. Scattering coefficients for different PBAPs vary by a factor of up to four and the absorption coefficients by a factor of up to six.

The difference of optical properties between bacteria species or fungi species can be larger than that between these two types of PBAPs. Therefore, detailed information on PBAP species is important in order to estimate their direct interaction with radiation (**Section 4.1.4**).

**c. Figure 7: why are the nitrated changes in scattering large at smaller wavelengths?**

**Response** (c): The scattering and absorption coefficients are affected by the real part and the imaginary part in non-linear ways. We modified discussion at line 453:

Due to the lack of data on the change of complex refractive index ( $\Delta m$ ) for nitrated proteins in PBAPs, we assume nitrated PBAPs have a similar change in the refractive index to that of SOA (Sopt12 and Sopt13). The scattering coefficient can change by up to 20% and the absorption coefficient by a factor of three at  $\lambda = 0.42 \ \mu m$  (*Figure 7*). After nitration, the scattering coefficient decreases by ~20% in the range of 300 nm

[revised manuscript text omitted]

*Minor comments:* 1. The acronym used in the paper is inconsistent with the literature on \*primary\* biological aerosol particles (PBAP) not BAP. While they do talk about some secondary processing of the aerosols, the origin of the particles is still primary (as opposed to secondary formation), and consistency with prior work is helpful.

**Responses 1:** We agree with the referee that terminology consistent with the literature should be preferred to avoid confusion. We have changed BAP to PBAP throughout the manuscript, including figures.

2. Line 57 – is the Londahl et al. 2014 the correct reference here? This seems to be an error.

**Responses 2:** We apologize for the confusion. It was indeed a wrong reference. We replaced it by the correct reference by the same author

3. Line 58 – Myhre et al. 2013 is not in the reference list.

**Responses 3:** The reference was added to the reference list.

**4. Line 98 – Pollen can also nucleate ice – see Diehl et al. 2001**

**Responses 4:** We already included pollen in *Table 1*. We discussed the ice nucleation property of pollen and modified the following sentences:

In addition to acting as CCN, some species of bacteria, fungi, and pollen can nucleate ice at high temperatures (Hoose and Möhler, 2012; Morris et al., 2004, 2008; Pouzet et al., 2017; Diehl et al., 2001, 2002), which makes them unique in terms of ice nucleation to affect the evolution of mixed-phase clouds at these temperatures (*Figure 1c*).

5. Line 181 – missing word between "that" and "might"? Can't tell what this sentence is supposed to say

**Responses:** Thanks for pointing out this omission. We completed the sentence as follows:

Due to the similarity of the molecular structure of organic macromolecules (e.g. proteins) and secondary organic aerosols (SOA), it can be likely assumed that nitration might alter the BAP refractive index similar to that of SOA.

6. Table 1: missing many references on the rupture of pollen. I actually think that these numbers are incorrect and very much mis-represent the range of potential sizes (see refs Grote, Taylor, Suphioulglu for a few; listed below). Also, you are missing the rupture of fungal spores (China, Lawler; see references below). Also missing the fact that the hygroscopicity of pollen may change on rupture (not just from oxidation).

**Responses 6:** We extended Table 1:

| BAP                             | Physicochemical properties             |                                                           |                                                                                                                                         |                     |                                                  |                                                               |                                                       |  |  |  |
|---------------------------------|----------------------------------------|-----------------------------------------------------------|-----------------------------------------------------------------------------------------------------------------------------------------|---------------------|--------------------------------------------------|---------------------------------------------------------------|-------------------------------------------------------|--|--|--|
|                                 | Concentration
N (cm -3 ) | Diameter
D (µm)                                        | Complex
refractive index
m $(\lambda) =$
n + ik                                                                                | Hygroscopicity
κ | Surface
tension
σ (mN
m -1 ) | Number
fraction of
PBAPs with
IN active
molecules | Contact
angle
θ(°)                              |  |  |  |
| Bacteria                        | 0.001-1 (1)                            | 1 (17);
0.6-7 (18)                                     | n: 1.5-1.56,
k: 3·10 -5 -6·10 -4
(24);
n: 1.5-1.56,
k: 0-0.04 (25); n:
1.25-1.85,
k: 0-0.5 (26) | 0.11-0.25 (27)      | 25, 30,
55, 72
(35)                        | ~0.1%, ~1%,
~10% (36)                                      | 32-34
(39);
4-20
(40);
28, 33,
44 (41) |  |  |  |
| Fungal spores                   | 0.001-0.01 (2)                         | 3-5 (4);
1-30 (5)                                      | n: 1.25-1.75,
k: 0-0.32 (26)                                                                                                         |                     |                                                  |                                                               | 30-33
(42)                                         |  |  |  |
| Subfungi
particles
(SFPs) | 150-455 (3)                            | 0.01-0.1
(3);
0.02-0.05
(3);
0.03-0.9
(19) |                                                                                                                                         |                     |                                                  |                                                               |                                                       |  |  |  |

**Table 1.** Physicochemical properties of various PBAPs and their changes due to physical, chemical and biological ageing processes based on literature data.

| Fern                             | 10-5 (4)                                                                                                                                           | 1-30 (4)                                          |                                                                                                                                                                                     |                                                                        |                                                                                                                                                                                                                                                                                                                                                                                                     |                  |                                    |
|----------------------------------|----------------------------------------------------------------------------------------------------------------------------------------------------|---------------------------------------------------|-------------------------------------------------------------------------------------------------------------------------------------------------------------------------------------|------------------------------------------------------------------------|-----------------------------------------------------------------------------------------------------------------------------------------------------------------------------------------------------------------------------------------------------------------------------------------------------------------------------------------------------------------------------------------------------|------------------|------------------------------------|
| spores                           |                                                                                                                                                    |                                                   |                                                                                                                                                                                     |                                                                        |                                                                                                                                                                                                                                                                                                                                                                                                     |                  |                                    |
| Pollen                           | 0.001 (5)                                                                                                                                          | 5-100
(20)                                     | n: 1.3-1.75,
k: 0.01-0.2 (26)                                                                                                                                                    | 0.03-0.073
(28);
0.036-0.04
(29);
0.05-0.1
0.08-0.17 (2 | 8
(30);
31)                                                                                                                                                                                                                                                                                                                                                                                   | ~100%
(37,38) | 14-30
(40);
15, 16.3
(43) |
| Subpollen
particles
(SPPs) | 0.1 (6)                                                                                                                                            | 1-4 (6);
0.03-4
(21);
0.12-4.67
(22); |                                                                                                                                                                                     | 0.14-0.24 (2
0.12-0.13 (2
0.1-0.2 (34)                           | 32);
33);
)                                                                                                                                                                                                                                                                                                                                                                                   |                  |                                    |
| Viruses                          | 0.01 (4)                                                                                                                                           | 0.01-0.3
(4)
0.04-0.2
(23)               |                                                                                                                                                                                     |                                                                        |                                                                                                                                                                                                                                                                                                                                                                                                     |                  |                                    |
| Ambient
PBAPs                 | 0.1-1 (7);
1-8 (8);                                                                                                                             | > 0.4 (7,8)                                       |                                                                                                                                                                                     |                                                                        |                                                                                                                                                                                                                                                                                                                                                                                                     |                  |                                    |
| Ambient
PBAPs                 | 0.2-1.2 (9);
0.04-0.13 (10);
0.012-0.095
(11);
0.01-1.4 (12);
0.57-3.3 (13);
0.1-0.43 (14);
0.02-0.09 (15);
0.005-0.5 (16) | > 1 (9-16)                                        |                                                                                                                                                                                     |                                                                        |                                                                                                                                                                                                                                                                                                                                                                                                     |                  |                                    |
| Ageing pro                       | cesses of PBAPs                                                                                                                                    |                                                   |                                                                                                                                                                                     |                                                                        |                                                                                                                                                                                                                                                                                                                                                                                                     |                  |                                    |
| Bacteria                         | Physical ageing
Agglomeration: $\Delta D > 0$ ,
$\Delta N < 0$ (18)
Rupture: $\Delta D < 0$ ,
$\Delta N > 0$ (3.19)                    |                                                   | Chemical ageing
Nitration: $\Delta n > 0$ , $\Delta k > 0$
(44) ;
Nitration: $\Delta \theta \sim 1^{\circ}$ (41);
pH changes:
$\Delta \theta \sim 1.5^{\circ}$ (41). |                                                                        | Biological ageingBiosurfactant production: $\sigma < 0$ (35);Biofilm formation: $\Delta D > 0$ (45);Endospore formation: $\Delta N > 0$ (46);Cell generation: $\Delta D > 0$ (47);Desiccation: $\Delta D < 0$ (48);Pigment formation: $\Delta k > 0$ (49,50);IN protein expression: $\Delta \theta < 0$ (no data yet)Biosurfactant production: $\sigma < 0$ (35);Germination: $\Delta N > 0$ (49) : |                  |                                    |
| Pollon                           |                                                                                                                                                    |                                                   | Oridation 0.5                                                                                                                                                                       |                                                                 | Desiccation: $\Delta$ (48).                                                                                                                                                                                                                                                                                                                                                                         | D < 0            |                                    |
| ronen                            | $\Delta N > 0$ (6,21,22)                                                                                                                           | )                                                 | Oxidation: $0.5 \leq (43)$                                                                                                                                                          | $\nabla A \ge 0.8^{\circ}$                                             |                                                                                                                                                                                                                                                                                                                                                                                                     |                  |                                    |

(1) Total bacteria, Tong and Lighthart et al., 2000; (2) Elbert et al., 2007; (3) After rainfall, Lawler et al., 2020; (4) Després et al., 2012; (5) blooming times, Huffman et al. 2010; (6) thunderstorm times, Zhang et al., 2019; (7) Based on protein dyes, Lake Baikal, Russia, Jaenicke, 2005; (8) Based on protein dyes, Mainz, Germany, Jaenicke, 2005; (9) In the Amazon, Whitehead et al., 2016; (10) In the Amazon, Huffman et al., 2012; (11) Puy de Dôme, Gabey et al. 2013; (12) In megacity Beijing, China, Wei et al., 2016; (13) In Megacity Nanjing, China, Yu et al., 2016; (14) High altitude, Ziemba et al., 2016b; (15) High altitude, Perring et al. 2015; (16) High concentration observed during and after rain, Huffman et al., 2013; (9) to (16) are based on autofluorescence of PBAPs; (17) Burrows et al., 2009a; (18) Lighthart 1997; (19) China et al., 2016; (20) Pöhlker et al., 2013; (21) Taylor et al., 2004; (22) Taylor et al., 2002; (23) Verreault et al., 2008; (24) Arakawa et al., 2003; (25) Thrush et al., 2010; (26) Hu et al. 2019; (27) Lee et al., 2002; (28) Pope et al. 2010; (29) Tang et al., 2019; (30) Chen et al., 2019; (31) Griffiths et al., 2012; (32) pollenkitt, Prisle et al., 2019; (33) Mikhailov et al., 2019; (34) Mikhailov et al., 2020; (35) Renard et al., 2016; (36) T ~-10 °C, immersion freezing,

*Pseudomonas syringae* bacteria, *Pseudoxanthomonas* sp., *Xanthomonas* sp., Joly et al., 2013; (37) deposition freezing for pollen, Diehl et al., 2001; (38) immersion and contact freezing for pollen, Diehl et al., 2002; (39) Hoose and Möhler, 2012; (40) Chen et al., 2008; (41) immersion freezing for *Pseudomonas syringae*, and *Pseudomonas fluorescens*, Attard et al., 2012; (42) immersion freezing for fungi, Kunert et al., 2019; (43) deposition freezing of silver birch and grey alder pollen, Gute and Abbatt, 2018; (44) nitrated SOA (toluene as precursor) to represent nitrated BAP, Liu et al., 2015; (45) Morris et al., 2008; (46) Enguita et al., 2003; (47) Ervens and Amato, 2020; (48) Barnard et al., 2013; (49) Pšenčík et al., 2004; (50) Fong et al., 2001.

**We added some texts in section 2.3.1 at line 225:**

The hygroscopicity of pollen is similar to that of bacteria: The  $\kappa$  value of intact pollen grains falls into the range of  $0.03 \le \kappa_{pollen} \le 0.17$  (Chen et al., 2019; Pope, 2010; Tang et al., 2019), pollenkitts (which are parts of pollen surface) and SPPs (which are fragments after rupture) are slightly more hygroscopic ( $0.14 \le \kappa_{pollenkitt} \le 0.24, 0.1 \le \kappa_{SPP} \le 0.2$ ) (Mikhailov et al., 2019; Prisle et al., 2019; Mikhailov et al., 2020) than intact pollen grains, which can be explained by the nonuniform composition of pollen (Campos et al., 2008).

In addition, in the conclusion section at line 720, we modified the following sentences:

 $\kappa_{PBAP}$  might be modified by physical (e.g., release of inner molecules due to rupture of pollen and fungal spores, condensation of gases), biological (e.g., formation of biosurfactants or other metabolic products), and chemical (e.g., nitration, oxidation) processes.

7. Lines 283-284: The text should more clearly state that certain classes of PBAP are excluded based on the 0.5-2.8 micron size representation.

**Responses 7:** We add at line 312:

Thus, the simulations focus on PBAPs in this size range and exclude smaller (e.g. viruses, SFPs or SPPs) and larger (e.g. pollen grains) particles.

8. Line 342: Fungal spores could also be on the order of this size. . .

**Responses 8:** We added at line 365:

Larger PBAPs ( $D_{PBAP} = 3 \ \mu m$ ,  $S_{opt6}$ ) such as SPPs and fungal spores lead to an increase in the scattering coefficient by a factor of 1.4-4.7 depending on  $\lambda$ .

9. Lines 360-363: This line downplays the potential importance of non-spherical particles. The true atmospheric range of moisture conditions is not enough to say what is more likely, therefore this speculation should be removed and it would be better to discuss what types of uncertainties non-spherical particles would include.

**Responses 9:** We removed the speculation at lines 360-363. We found some papers about the uncertainties of non-spherical particles. We added the following to the end of section 4.1.1 at line 384:

Non-sphericity of particles might translate into the same changes as caused by different particles sizes, which might induce uncertainties including optical depth and surface albedo (Kahnert et al., 2007). These uncertainties on scattering and absorption caused by non-spherical shape might be of comparable magnitude to that caused by the complex refractive index (Yi et al., 2011).

10. Figure 3: the caption states that there is an a/b panel to capture scattering and absorption, yet only the scattering is shown.

**Responses 10:** We apologize for the omission of Figure 3b in the original manuscript. Actually, we added Figure 3b in the supporting information (as Figure S2) We changed the Figure caption accordingly. Its information is rather limited since the absorption for all PBAPs is (nearly) identical, i.e., the absorption coefficient is not affected in the presence of PBAPs.

11. Line 392: "very small PBAP" could also be pollen or fungal fragments. Please see literature suggestions in the major comments.

**Responses 11:** We have changed the sentences at line 427:

Only for very small PBAPs, i.e. representative for viruses, SPPs or SFPs (*Section 2.1*), the curvature term significantly influences *s* (*Figure 5*).

12. Line 422: I think this is supposed to be \$15 and \$16?

**Responses 12:** The referee is correct; we meant to refer to S15 and S16, rather than S13 and S14. As we deleted some simulations in Table 3 to make it shorter, their number changed to S12 and S13.

**13. Line 471: what is delta\_mBAP? First use, please define. (perhaps including S13, S15 and S16?)**

**Responses 13:** Delta-mBAP means the change of refractive index due to different types of BAP or nitration. We will define Delta-mBAP at line 454. S13 means simulation 13. We will define them at first use at line 316, line 331, and line 343.

14. Table 3: last row – is dm\_aged the same as dm\_nitrated? Different terminology than Table 2.Responses 14: They are the same. We use now dm\_nitrated for the whole manuscript.

15. Also in Table 3 – what is the dm actually referring to? Hard to tell from comparing with Table 2.

**Responses:** dm means the change of refractive index. We defined this at line 454. We also included more information in Table 3 (see response to Comment 3b).

**16. Lines 495-298: Could be compared with the observed values of Sc from Steiner et al. 2015**

**Responses:**